# Weight Space Correlation Analysis: Quantifying Feature Utilization in Deep Learning Models

**Chun Kit Wong**[*,1,2]                                                     CKWO@DTU.DK
**Paraskevas Pegios**[*,1,2]                                                  PPAR@DTU.DK
**Nina Weng**[1]                                                            NINWE@DTU.DK
**Emilie Pi Fogtmann Sejer**[3,4]                   EMILIE.PI.FOGTMANN.SEJER.01@REGIONH.DK
**Martin Grønnebæk Tolsgaard**[3,4]              MARTIN.GROENNEBAEK.TOLSGAARD@REGIONH.DK
**Anders Nymark Christensen**[1]                                            ANYM@DTU.DK
**Aasa Feragen**[1,2]                                                       AFHAR@DTU.DK

[1] *Technical University of Denmark, Kongens Lyngby, Denmark*

[2] *Pioneer Centre for AI, Copenhagen, Denmark*

[3] *University of Copenhagen, Copenhagen, Denmark*

[4] *CAMES Rigshospitalet, Copenhagen, Denmark*

**Editors:** Accepted for publication at MIDL 2026

## Abstract

Deep learning models in medical imaging are susceptible to shortcut learning, relying on confounding metadata (e.g. scanner model) that is often encoded in image embeddings. The crucial question is whether the model actively utilizes this encoded information for its final prediction. We introduce Weight Space Correlation analysis, an interpretable methodology that quantifies feature utilization by measuring the alignment between the classification heads of a primary clinical task and auxiliary metadata tasks. We first validate our method by successfully detecting artificially induced shortcut learning. We then apply it to probe the feature utilization of an SA-SonoNet model trained for Spontaneous Preterm Birth (sPTB) prediction. Our analysis confirmed that while the embeddings contain substantial metadata, the sPTB classifier's weight vectors were highly correlated with clinically relevant factors (e.g. cervical length) but decoupled from clinically irrelevant acquisition factors (e.g. scanner). Our methodology provides a tool for verifying model trustworthiness, by inspecting whether it utilizes features unrelated to the genuine clinical signal. Code available at https://github.com/wong-ck/wsc-analysis.

**Keywords:** Shortcut learning, feature utilization, obstetric ultrasound.

## 1. Introduction

Deep learning models have achieved impressive performance across numerous medical imaging tasks, often matching or exceeding human expert capabilities (De Fauw et al., 2018; Esteva et al., 2017). However, their reliance on vast, complex datasets introduces significant concerns regarding model trustworthiness. One of the primary threats to model reliability is shortcut learning, where a model learns a simple, non-causal predictive rule that performs well on training data but fails when deployed in new environments (Geirhos et al., 2020; Neuhaus et al., 2023; Jabbour et al., 2020). In medical imaging, this often manifests as

---

[*] Contributed equally

a model relying on confounding factors or shortcuts, which are variables correlated with both the image features and the clinical outcome, but without a direct causal link to the underlying anatomy or pathology, such as scanner models or acquisition protocols.

There is an ever-present possibility that non-disease related features, such as patient demographics, types of scanners or pre-processing protocols are subtly encoded within a medical image's visual features for deep learning models; as unlike human experts, machines are inherently better at capturing textures or minute subtleties. Gichoya et al. (2022) demonstrated that deep learning classifiers can predict a patient's self-reported race from chest X-ray images with high accuracy. This confirmed that models can extract non-clinical demographic information embedded in images, raising serious questions about fairness and generalization across different populations and settings (Obermeyer et al., 2019).

Although Gichoya et al. (2022) demonstrated that confounding information, such as race, is encoded in medical images, this is not equivalent to the model making use of that information during inference. We argue that the mere presence of a feature in the embedding space is a prerequisite for shortcut learning, but not a proof of its use. Glocker et al. (2023) addressed this gap by proposing a method to discover patterns and clustering within embeddings of the training data, and quantifying the separation of embeddings relative to ground-truth metadata labels using the Kolmogorov-Smirnov (K-S) test. Stanley et al. (2025) further showed that such confounders can be encoded across different layers of the model, and yet it does not necessarily imply that the model relies on them for its prediction. While these allow one to examine the distribution of the images in the embedding space, it still does not directly address the fundamental question: *Does the model's final classification layer actively leverage the embedded features of the confounder?* Another branch of work evaluates shortcut learning by generating counterfactual test sets that manipulate suspected shortcut features (Kumar et al., 2023; Fathi et al., 2024; Weng et al., 2024; Bender et al., 2025; Bender and Morik, 2026). While these can be valuable for highly localized artifacts such support medical devices, it might be hard to adapt them across metadata factors such as scanner type or non-localized patient characteristics.

To definitively address the question of information utilization, we introduce a methodology that focuses on comparing the attention of the neurons in the classification head quantitatively. Our method, which we termed Weight Space Correlation (WSC) analysis, distinguishes itself by comparing the weight vectors (i.e. the "attention") of the primary task against the weight vectors of auxiliary metadata tasks. By quantifying the alignment between these decision boundaries using correlation, we can directly determine whether the features used for a clinical prediction are the same features used for a confounder.

We motivate and evaluate our method in the context of fetal ultrasound, where models can inadvertently rely on shortcuts. Ultrasound images often contain on-screen annotations such as calipers and text (Mikolaj et al., 2023; Lin et al., 2024b), which can act as shortcuts in applications including anatomical classification and standard plane quality assessment (Baumgartner et al., 2017; Lin et al., 2024a; Wiśniewski et al., 2025; Pegios et al., 2025), out-of-distribution detection (Olsen et al., 2024; Wong et al., 2025) as well as spontaneous preterm birth prediction (sPTB) (Włodarczyk et al., 2020, 2019; Pegios et al., 2023). In this work, we focus on patient characteristics and other metadata that can also act as confounders in medical imaging because models may encode and exploit non-causal signals correlated with outcomes (Gichoya et al., 2022). In fetal ultrasound, related studies (Fournel

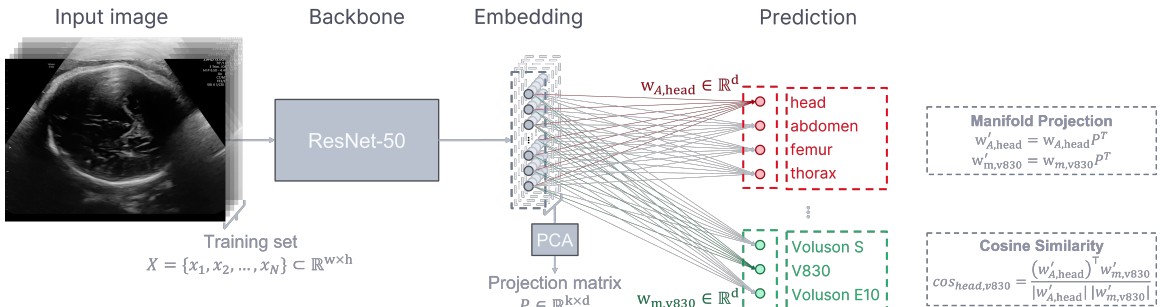

Figure 1: The WSC analysis workflow. An image is passed through the backbone encoder to produce an embedding, which is used by both the primary task (e.g. anatomy) and the auxiliary/metadata (e.g. scanner) classification heads. The weights of these heads are projected into a lower-dimensional space via PCA and their cosine similarity is computed to quantify the correlation, indicating feature utilization.

et al., 2025; Sejer et al., 2025) have mainly assessed bias through stratified performance analyses across patient metadata groups and imaging-related factors for sPTB deep learning models such as SA-SonoNet (Pegios et al., 2023). In contrast, in this work, we go beyond group-wise performance differences and directly *quantify feature utilization* by measuring the alignment between the primary clinical head and auxiliary metadata heads.

## 2. Method

Our goal is to determine whether a clinical prediction task implicitly relies on metadata-related information learned during training. Our method consists of three steps: (i) representing each task through the linear decision directions of its classification head, (ii) projecting these directions onto the intrinsic data manifold, and (iii) quantifying the reliance between tasks via cosine similarity of their projected weight vectors.

### 2.1. Latent Representation and Linear Classification Heads

We consider deep image classifiers in which the backbone encoder produces a final-layer latent representation that serves as input to a linear classification head. An input image $x$ is mapped to a feature vector $z = f_\theta(x) \in \mathbb{R}^d$, and a task-specific linear layer maps this representation to class logits $\ell = Wz + b$, where $W \in \mathbb{R}^{C \times d}$ is the matrix of classifier weights whose rows $w_i^\top$ correspond to class-specific weight vectors, and $b \in \mathbb{R}^C$ is the vector of class-specific bias terms. The predicted probabilities are given by $\hat{y} = \mathrm{softmax}(\ell)$. Each weight vector $w_i$ specifies how evidence for class $i$ changes as the embedding moves in latent space. Intuitively, $w_i$ indicates the feature direction that most increases the model's confidence in class $i$. For any task $t$ with $C_t$ classes, we denote its classifier parameters by $W_t \in \mathbb{R}^{C_t \times d}$ and $b_t \in \mathbb{R}^{C_t}$. We conceptualize the classifier weights not merely as regression coefficients, but as attention vectors acting upon the latent embedding. If the weights for the primary task are highly correlated with the weights for metadata attributes, it suggests the model attends to similar features for both predictions, implying a reliance on that specific shortcut.

## 2.2. Projection onto the Data Manifold

To compare feature utilization across tasks, we express classifier weights in a shared low-dimensional coordinate system derived from the data manifold via Principal Component Analysis (PCA). More specifically, let $Z = f_\theta(X) \in \mathbb{R}^{N \times d}$ denote the (zero-mean) latent embeddings of the training set of size $N$. We compute the empirical covariance of $Z$ and extract the top $k$ principal components explaining 99% of the variance in the dataset, while enforcing a minimum floor of 50 components (see Section 3.5). These form the projection matrix $P \in \mathbb{R}^{k \times d}$. Then, each classifier head is projected into this subspace:

$$W_t' = W_t P^\top, \qquad W_t' \in \mathbb{R}^{C_t \times k} \tag{1}$$

where the $i$-th row $w_{t,i}'$ denotes the class-specific decision direction after projection. If the latent representation satisfies $z \approx P^\top z'$, then $W_t z \approx W_t P^\top z' = W_t' z'$. This step ensures that the correlation is calculated based on the directions of variance that actually exist in the data, rather than the less informative orthogonal dimensions.

## 2.3. Quantifying Shortcut Reliance via Weight Space Correlation

Given a primary clinical task $A$ and task $m$ related to metadata information we assess whether two tasks rely on similar latent directions, using the cosine similarity between their projected class-specific weight vectors. Using Equation (1), for tasks $A$ and $m$, we have $W_A' = W_A P^\top$ and $W_m' = W_m P^\top$. Cosine similarity between class $i$ of task $A$ and class $j$ of task $m$ is defined as:

$$\cos_{ij} = \frac{(w_{A,i}')^\top w_{m,j}'}{\|w_{A,i}'\| \, \|w_{m,j}'\|}, \qquad i = 1, \ldots, C_A, \; j = 1, \ldots, C_m. \tag{2}$$

This yields the task-pair correlation matrix $\mathrm{Corr}(A, m) \in \mathbb{R}^{C_A \times C_m}$ which captures alignment between the decision directions of the two tasks within the intrinsic data manifold. High correlation indicates reliance on similar latent directions and may signal shortcut usage when $m$ corresponds to metadata. A high correlation in this projected space serves as a quantitative proxy for shortcut learning: it implies that the decision boundary for the clinical task aligns closely with the decision boundary for the shortcut.

## 3. Experiments and Results

### 3.1. Clinical Datasets

We utilize two distinct, private clinical ultrasound datasets to evaluate the interplay between feature encodability and shortcut learning. Both datasets are accompanied by a rich set of demographic and acquisition metadata, including ultrasound scanner manufacturer, hospital site ID, and maternal ethnicity.

- **The Fetal Dataset**: This dataset focuses on anatomical classification. It comprises 2D ultrasound images of four standard fetal planes: the fetal head, abdomen, femur, and thorax. The primary task is a multi-class classification problem where the model must identify the anatomical plane present in the image (Sendra-Balcells et al., 2023).

- **The Cervix Dataset**: This dataset focuses on a prognostic task related to preterm birth, defined as delivery before 37 weeks of gestation. It consists of transvaginal cervical ultrasound images, balanced equally between two classes: term birth and preterm birth. The primary task is the binary classification of the image into these prognostic outcomes (Pegios et al., 2023; Sejer et al., 2025).

**Preprocessing of Metadata Attributes:** To unify the prediction tasks under a single framework, we reformulated the prediction of continuous metadata variables as a multi-class classification problem. Continuous attributes were discretized via binning, transforming scalar values into distinct categorical labels. The range of each continuous variable was partitioned into $k$ intervals. Any value falling within a specific interval was assigned the class label corresponding to that bin. This discretization step mitigates the impact of outliers and allows for the application of classification metrics across all target variables.

### 3.2. Establishing the Embedding of Metadata in Images

The foundational question addressed in our experiments is whether the metadata attributes, both clinical and acquisition-related, are implicitly embedded within the visual features of the medical images themselves. To test this, we trained standard ResNet50 models as our **baseline** models to predict metadata factors directly from images in both fetal and cervix datasets (see Section 3.1). We trained separate classification models for each metadata factor in each dataset. The performance of these baseline models, which simply predict a single metadata factor from the raw image input, is documented under Appendix A.

The results consistently demonstrate that the visual features extracted by the model contain substantial information regarding the metadata across both domains. In the fetal dataset, models achieved strong predictive accuracy for the primary task, as well as for acquisition-related factors like scanner, pixel spacing, and hospital ID. Similar performance was also observed among models predicting auxiliary factors from the cervix images.

These findings confirm a consistent observation: for both datasets, some of the metadata factors are implicitly embedded within the image's texture, geometry, and presentation. This suggests that any model trained on these images may inadvertently encode features related to these attributes alongside the primary clinical task, necessitating the subsequent investigation into their utilization (see Section 3.3).

**Characterizing the Null Distribution of WSC Values:** Before analyzing specific metadata dependencies, we established a reference by determining the distribution of WSC values expected by chance or through architectural constraints. We aggregated all baseline models trained using the fetal dataset and extracted their classification heads. We then computed the pairwise cosine similarity between all combinations of weight vectors, categorizing them into two distinct groups to produce the histograms:

- Intra-Task Correlation: Pairs of weight vectors taken from the same classification head. These represent the alignment between different classes within a single task (e.g. the weight vector for "Fetal Head" vs. "Fetal Abdomen").

- Inter-Task Correlation (Null Distribution): Pairs of weight vectors taken from different classification heads (e.g. the weight vector for "Fetal Head" vs. "Scanner Model B").

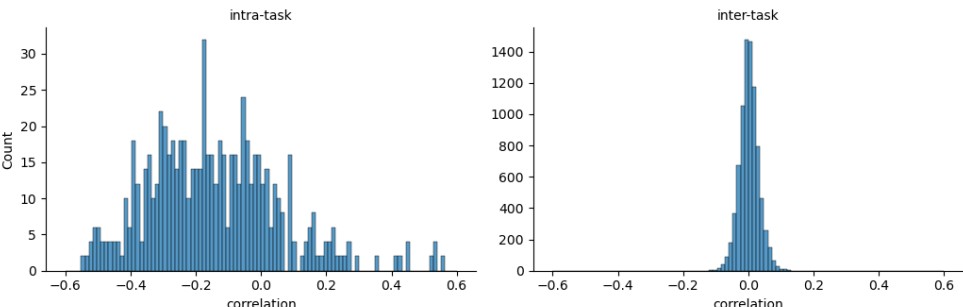

Figure 2: Reference distributions of WSC values. (Left) The intra-task correlation between different classes within the same head. (Right) The null distribution of correlations between weight vectors from unrelated classification tasks.

As shown in Figure 2, the inter-task correlation values form a narrow, zero-centered distribution. This "null distribution" confirms that under typical conditions, the decision boundaries for unrelated tasks are nearly orthogonal in the projected weight space. In contrast, the intra-task correlation exhibits a slight negative bias, reflecting the competition between classes in a softmax-based multiclass objective, where the model must learn to distinguish between mutually exclusive categories. This null distribution enables us to more confidently identify shortcut behavior in downstream experiments: any inter-task WSC value that significantly deviates from this zero-centered baseline provides quantitative evidence of shared feature utilization.

### 3.3. Utilization of Clinically Irrelevant Factors in Classification

This section addresses our second core research question: Does the primary classifier (fetal plane identification) actively utilize these clinically irrelevant, but embedded, factors in its decision-making process? To answer this, we established a baseline for "encodability", i.e. the degree to which metadata is present in the latent space regardless of its utility, we adopt the linear probing methodology described by Gichoya et al. (2022); Glocker et al. (2023).

We first trained baseline models for the primary clinical task of fetal standard plane classification. We then freeze the parameters of the backbone encoder, treating it as a fixed feature extractor. We discard the primary classification head and attach new, randomly initialized fully connected heads corresponding to the metadata attributes. These probing heads are then trained to predict the metadata (e.g. scanner, hospital ID) using only the frozen embeddings. High performance on this probing task indicates that the model has encoded information about the metadata, even if it was not explicitly trained to do so.

The performance of the fine-tuned model on the metadata prediction tasks, shown in Table 1, confirms the continued presence of this information in the embeddings of the primary classifier. The model achieved a relatively high AUROC when fine-tuned to predict certain metadata variables (e.g. scanner). This result reinforces the finding that the image embeddings, generated by a model focused solely on plane classification, still contain sufficient features to distinguish between different acquisition parameters.

| | Fine-tuned model | | | Multitask model | | |
|---|---|---|---|---|---|---|
| **Target** | **Accuracy** | **F1** | **AUROC** | **Accuracy** | **F1** | **AUROC** |
| Plane | $95.8 \pm 0.6$ | $95.4 \pm 0.6$ | $99.6 \pm 0.1$ | $95.7 \pm 0.1$ | $95.3 \pm 0.1$ | $99.6 \pm 0.1$ |
| Scanner | $76.8 \pm 1.4$ | $75.3 \pm 1.7$ | $91.4 \pm 0.9$ | $96.9 \pm 0.5$ | $96.7 \pm 0.6$ | $99.7 \pm 0.1$ |
| Pixel Spacing | $56.8 \pm 1.1$ | $52.9 \pm 1.3$ | $88.2 \pm 0.3$ | $69.4 \pm 1.2$ | $67.1 \pm 1.2$ | $94.0 \pm 0.4$ |
| GA | $50.5 \pm 1.7$ | $44.4 \pm 1.2$ | $80.0 \pm 1.0$ | $59.5 \pm 1.3$ | $51.6 \pm 1.0$ | $86.8 \pm 0.4$ |
| Hospital ID | $49.8 \pm 3.4$ | $35.0 \pm 2.2$ | $77.0 \pm 1.1$ | $73.7 \pm 0.7$ | $54.5 \pm 1.7$ | $91.9 \pm 0.7$ |
| Year Of Study | $46.4 \pm 3.2$ | $39.4 \pm 0.8$ | $75.6 \pm 1.9$ | $68.7 \pm 1.1$ | $49.9 \pm 1.0$ | $90.5 \pm 0.2$ |
| BMI | $36.5 \pm 1.6$ | $35.2 \pm 1.5$ | $64.1 \pm 1.1$ | $39.7 \pm 1.8$ | $40.0 \pm 1.8$ | $68.6 \pm 0.9$ |
| Ethnicity | $85.9 \pm 5.8$ | $50.0 \pm 1.6$ | $58.8 \pm 5.8$ | $94.3 \pm 0.3$ | $48.5 \pm 0.1$ | $42.0 \pm 2.3$ |
| Term Birth | $74.3 \pm 2.9$ | $53.4 \pm 2.7$ | $57.1 \pm 3.6$ | $81.8 \pm 0.3$ | $50.3 \pm 2.1$ | $56.9 \pm 2.6$ |
| Parity | $49.7 \pm 3.2$ | $33.9 \pm 0.3$ | $55.0 \pm 1.0$ | $70.2 \pm 2.1$ | $32.1 \pm 0.6$ | $64.0 \pm 1.5$ |
| Smoking | $77.2 \pm 7.0$ | $49.5 \pm 1.9$ | $50.8 \pm 3.9$ | $84.5 \pm 0.4$ | $48.2 \pm 1.3$ | $51.2 \pm 2.7$ |
| Maternal Age | $25.6 \pm 1.8$ | $24.6 \pm 1.4$ | $50.8 \pm 1.7$ | $28.3 \pm 1.3$ | $25.0 \pm 2.7$ | $53.3 \pm 1.6$ |

Table 1: Test performance of ResNet50 classifier model when fine-tuned to predict the other targets, or trained to predict various targets in a multitask setting.

While the embeddings hold the information, the critical step is, however, determining if that information is being used. As demonstrated in Figure 3(a), WSC analysis on the weight vectors of the final classification heads suggests that the correlation between the weight vectors for the fetal plane classes and the weight vectors for the scanner classes was consistently low. This indicates that although the necessary information about the scanner is present in the preceding embedding layer, the model's decision boundary for the primary plane classification task is largely orthogonal to the directionality required to classify the scanner. In other words, the embedded, clinically irrelevant information is not being actively utilized by the classifier for its primary prediction.

To stress-test this finding, we trained a *multi-task learning* model designed to encourage the simultaneous encoding of all factors. This model extends ResNet50 with auxiliary output heads, which was trained to predict the fetal plane and all metadata factors concurrently, thereby explicitly maximizing the metadata information content within the shared embedding space, as shown in Table 1. Upon performing the same analysis on this model, the correlation between the plane and scanner weight vectors remained low. This further reinforces the initial conclusion: even when the model is explicitly forced to encode metadata information into the embeddings, the weight vectors for the primary plane classification task remain largely decoupled from the weight vectors of the irrelevant scanner factors.

### 3.4. Validation of Shortcut Learning Detection via Induced Bias

Section 3.3 suggested that the baseline models did not utilize available scanner information for plane classification. This section aims to validate our hypothesis: if a model does adopt a shortcut learning strategy, this behavior will be detectable via WSC analysis. Inspired by Weng et al. (2024), we derived two specific sub-datasets from the full fetal dataset:

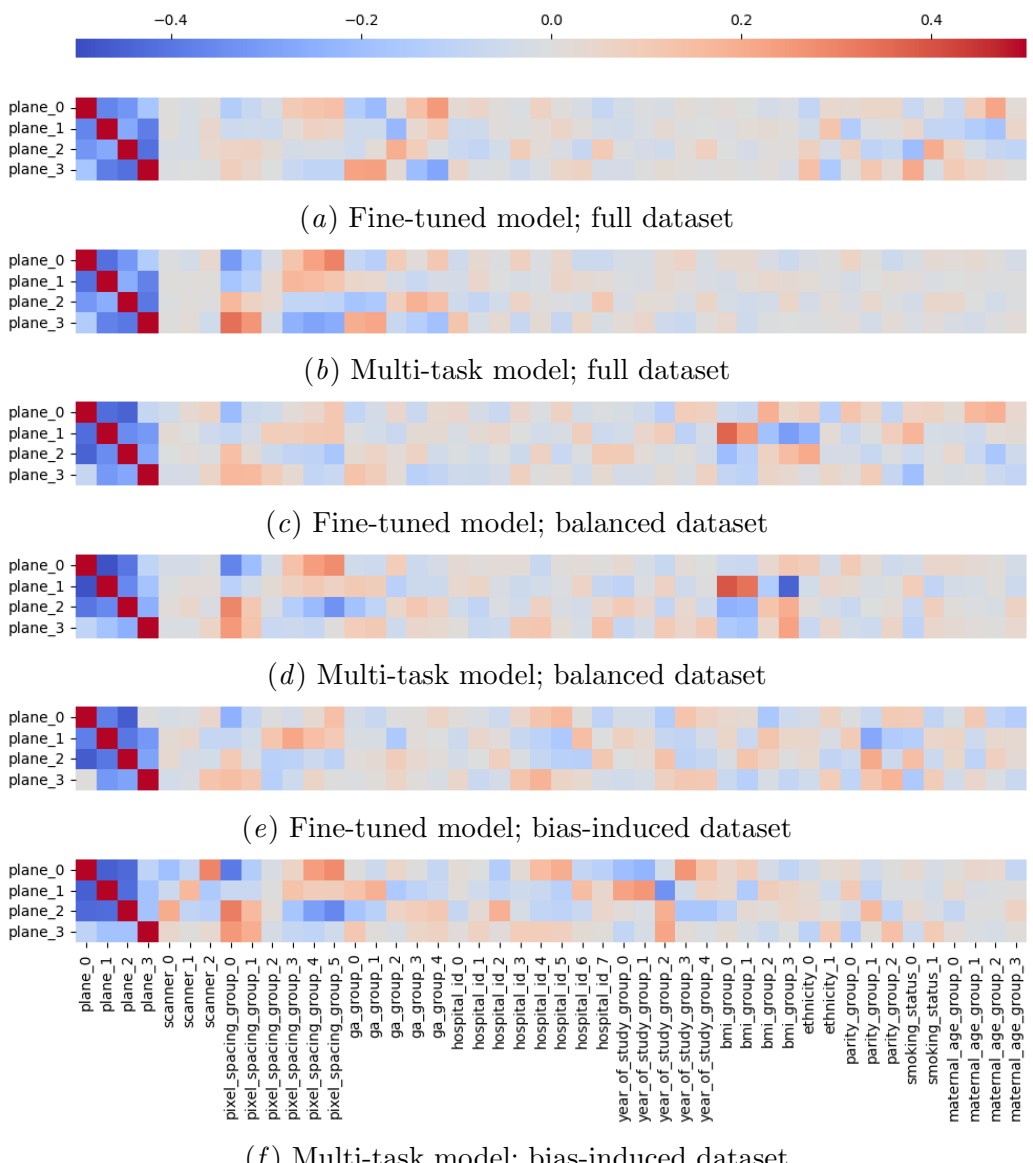

(a) Fine-tuned model; full dataset

(b) Multi-task model; full dataset

(c) Fine-tuned model; balanced dataset

(d) Multi-task model; balanced dataset

(e) Fine-tuned model; bias-induced dataset

(f) Multi-task model; bias-induced dataset

Figure 3: Correlation matrix between weight vectors from classification head of the primary task versus that of each metadata attributes, extracted from fine-tuned or multi-task classifier models, trained using the entire, balanced, or biased fetal dataset. Full matrix available under Appendix B.

- Balanced Dataset: We performed data culling to ensure that the number of images in each plane-scanner class pair was balanced. This dataset serves as a rigorous control, ensuring no correlation exists between the clinical target and the acquisition metadata.

- Induced Bias Dataset: We intentionally introduced a strong correlation between the primary classification target (fetal plane) and a clinically irrelevant factor (scanner).

| Plane | Full | | | Balanced | | | Biased | | |
|---|---|---|---|---|---|---|---|---|---|
| | Voluson S | V830 | E10 | Voluson S | V830 | E10 | Voluson S | V830 | E10 |
| Abdomen | 717 | 333 | 658 | 300 | 300 | 300 | 150 | 150 | 658 |
| Head | 1018 | 805 | 888 | 300 | 300 | 300 | 150 | 700 | 150 |
| Femur | 760 | 315 | 533 | 300 | 300 | 300 | 700 | 150 | 150 |
| Thorax | 855 | 602 | 523 | 300 | 300 | 300 | 700 | 150 | 150 |

Table 2: Composition of the full, balanced, and biased fetal plane dataset.

We discarded images such that the majority of images for each standard plane were acquired by a distinct, single scanner, forcing the model to potentially adopt a shortcut where recognizing the scanner acts as an efficient proxy for anatomical classification.

The composition of these datasets is documented in Table 2, illustrating the contrast between the balanced control and the high degree of induced correlation in the biased set.

We repeated the analysis described in Section 3.3 using these sub-datasets, training both single-task and multi-task classifiers on each. As detailed in Appendix C, the predictive performance for both plane and scanner classification remained consistent across the full, balanced, and biased datasets. However, this stability does not extend to the internal weight alignments, as shown in Figures 3(e) and 3(f). In the balanced scenario, the WSC values between plane and scanner weight vectors remained within the null distribution, confirming that the model did not associate these tasks when the data was uncorrelated. Meanwhile, in the induced bias scenario, the single-task classifiers showed an increase in WSC values. More crucially, the multi-task model exhibited a significantly stronger alignment between the weight vectors for fetal plane classes and scanner classes. This increase confirms that the model adopted the shortcut when bias was present, with the decision boundary for plane classification aligning with the directionality required for scanner classification.

While the strong WSC values observed between the plane and scanner weight vectors in the biased scenario is indicative of a dependency between the two prediction tasks, it is important to note the nature of this association. A high correlation coefficient signifies that the two tasks assign similar attention vectors to the shared model embeddings; they are looking at similar features in the embedding space to make their respective decisions. It does not explicitly define the direction of the shortcut. That is, the result does not prove whether the model is using scanner information to predict the plane, or if the plane information is strongly predictive of the scanner. It simply confirms that, under conditions of high dataset bias, the feature utilization for the two tasks becomes strongly coupled.

The findings from this experiment provide crucial validation. High WSC is a reliable indicator of shortcut learning, where the model utilizes a non-causal, highly correlated factor for its prediction. The successful detection of this induced shortcut proves that the WSC analysis is effective in determining the active utilization of embedded, irrelevant information.

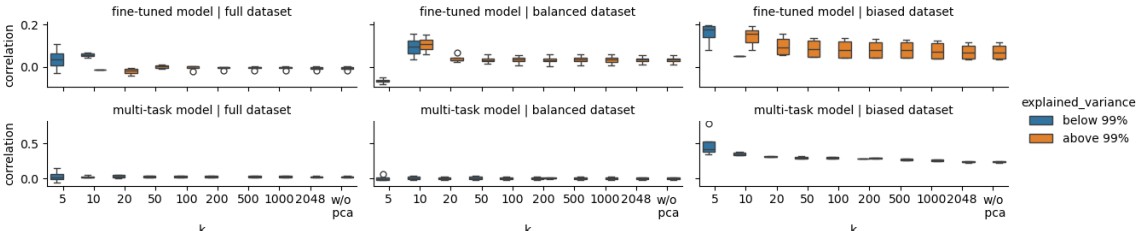

Figure 4: Sensitivity of WSC values to the number of principal components $k$. The WSC values are calculated using weight vectors for plane_0 and scanner_2, a pair representative of the general trends observed across all plane-scanner combinations (detailed in Appendix D).

### 3.5. Empirical Determination of the PCA Projection Threshold

After introducing the experiments in Sections 3.3 and 3.4, this section takes a detour to explain the empirical rationale behind the parameter choices made in our dimensionality reduction strategy (see Section 2.2). Specifically, we investigate the choice of $k$, the number of principal components used to define the data manifold projection.

To determine the optimal $k$, we conducted a sensitivity analysis across four random seeds. This was performed for both the fine-tuned baseline models and the multi-task models, using all versions of the fetal dataset (i.e. full, balanced, and biased). We performed WSC analysis between the fetal plane classification head and the scanner classification head, varying $k$ from 10 up to the full dimensionality of the ResNet50 embedding space ($d = 2048$).

The results of this analysis are shown in Figure 4, including a reference line for correlation calculated without projection (i.e. in the raw embedding space). Two trends are observed:

- Unstable Mean and High Variance at Low $k$: For small values of $k$, we observed high variance in correlation values across random seeds. Furthermore, at these low values, the mean correlation had not yet settled into the plateau it eventually reaches as $k$ increases. This indicates that a very low-dimensional projection is insufficient to capture the manifold and is sensitive to the stochasticity of individual training runs.

- Underestimation at High $k$: As $k$ approached the full dimension, the correlation values decreased slightly. This suggests that including all dimensions may dilute the meaningful alignment between tasks, underestimating the shortcut learning effect.

Based on these empirical observations, we established a strategy to balance stability and sensitivity. We set a floor value of $k = 50$ to ensure sufficient representational capacity and eliminate seed-based variance, while simultaneously requiring that the chosen $k$ must capture at least 99% of the variance in the latent embeddings.

### 3.6. Probing a Trained Model: Analysis of SA-SonoNet Embeddings

Having validated our WSC analysis methodology in Section 3.4, we now apply our technique to probe a model trained on a real-world, relevant clinical task. For this analysis, we utilize

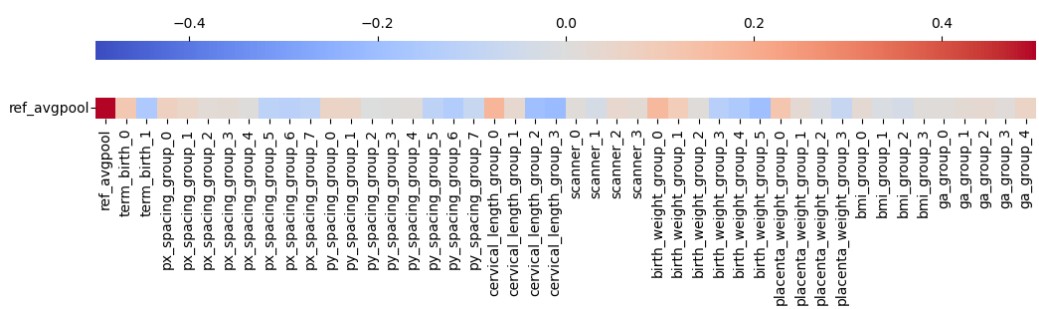

Figure 5: WSC values between weight vectors from classification head of the targets of the fine-tuned SA-SonoNet model. `ref_avgpool` represents a flattened average pooling layer, which is the original model's classification head. Full covariance matrix is available under Appendix B.

the SA-SonoNet model (Pegios et al., 2023), which achieved state-of-the-art performance on the challenging task of Spontaneous Preterm Birth (sPTB) prediction.

SA-SonoNet is a shape- and spatially-aware network based on the SonoNet (Baumgartner et al., 2017) architecture, modified to predict term or preterm birth from transvaginal cervix ultrasound images. The key innovation is its multimodal input: given cervix image, the model first leverages a segmentation network (Lin et al., 2023) to compute a segmentation map of important anatomical structures (e.g. cervical canal and boundaries). The final input to the SA-SonoNet classifier is the concatenation of the original image, the segmentation map, and the pixel spacing values, which are repeated and reshaped to image dimensions to inject spatial information. The original SA-SonoNet model uses an average pooling layer on a $14 \times 18$ 2D embedding feature map for its final prediction. We first flattened this $14 \times 18$ feature map into a 252-long 1D embedding vector. This vector was then connected to a newly initialized fully connected classification head for predicting the metadata variables. We modified its final classification layer for fine-tuning and analysis. For the WSC analysis, we represented the original average pooling operation, which maps the 252-long embedding to the final sPTB prediction, by including a 252-long vector of all ones in the set of weight vectors. This step allows us to compare the feature utilization direction of the original sPTB task against the fine-tuned metadata tasks.

The fine-tuned model's performance on various auxiliary metadata tasks and the resulting WSC values are presented in Table 4 under Appendix A, and the associated correlation matrix Figure 5. The analysis reveals several expected correlations, confirming that the model utilizes clinically significant factors, as well as some desirable decoupling from irrelevant factors. Cervical length showed a moderate correlation, aligning with its role as the clinical gold standard for sPTB risk. Pixel spacing also demonstrated a moderate correlation, reflecting its known influence on model performance and confirming that this imaging parameter contributes to the prediction. In contrast, scanner type showed only a weak correlation, suggesting the model does not rely on acquisition hardware–related shortcuts.

Meanwhile, it is worth noting that although birth weight exhibited a strong WSC value with sPTB prediction, which is superficially consistent with the expected clinical link be-

tween prematurity and low birthweight, this finding must be interpreted with caution. As shown in Table 4, the predictive performance for the birth weight probing head was low, indicating that this metadata is not effectively encoded in the model's embeddings. Consequently, the high WSC value in this specific instance is functionally meaningless.

## 4. Limitations and Discussion

Our method assumes that the relationship between the embedding space and the class predictions is linear. While this might seem restrictive, it aligns with standard deep learning architectural conventions where the backbone acts as a non-linear feature extractor, and the classification head functions as a linear probe on the resulting manifold. Furthermore, research suggests that the goal of supervised training is to linearize the data manifold, making classes linearly separable in the embedding space (Bengio et al., 2013).

It is also vital to distinguish between "shortcut learning" and other forms of model degradation. A model that exhibits low WSC values for a metadata attribute is not necessarily immune to failure when that attribute shifts. If a model performs poorly on a new scanner despite low weight alignment, our method suggests that the failure is not due to the scanner type being used as a shortcut. Instead, it may stem from other factors:

- **Data Quality:** Images from the new scanner may lack the diagnostic signal-to-noise ratio present in the training data.

- **Feature Shift:** The anatomical features learned by the model may be rendered differently by the new hardware, causing the embeddings to fall outside the distribution of the original decision boundary.

In such cases, our method rules out shortcut learning as the cause of a performance drop.

Finally, while we demonstrate successful detection of shortcuts based on individual metadata attributes, real-world shortcuts may be multifaceted, involving non-linear combinations of several confounding factors. Future work will investigate the extension of WSC analysis to higher-order interactions between multiple metadata dimensions.

## 5. Conclusion

In this work, we introduced Weight Space Correlation analysis, a simple and interpretable methodology designed to move beyond simply identifying the presence of confounding information to definitively quantifying its utilization by a deep learning classifier. We validated our method by successfully detecting artificially induced shortcut learning in a controlled environment. Applying this method to the SA-SonoNet model, we confirmed that while clinically irrelevant factors are indeed encoded in the image embeddings, the model's decision boundary for Spontaneous Preterm Birth prediction is selectively aligned with clinically meaningful metadata and, crucially, decoupled from confounding acquisition factors like scanner model. These findings provide a necessary level of trustworthiness in complex medical imaging models by confirming that the classifier is learning features relevant to the clinical task, rather than relying on spurious correlations.

## Acknowledgments

This work is funded by the Danish Pioneer Centre for AI (DNRF grant number P1), SONAI - a Danish Regions' AI Signature Project, and the Novo Nordisk Foundation through the Center for Basic Machine Learning Research in Life Science (NNF20OC0062606).

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

## Appendix A. Full Results: Establishing the Embedding of Metadata in Images

| Target | Accuracy | Precision | Recall | F1 | AUROC |
|---|---|---|---|---|---|
| **Fetal plane dataset** | | | | | |
| Plane | 95.8 ± 0.6 | 95.4 ± 0.6 | 95.4 ± 0.6 | 95.4 ± 0.6 | 99.6 ± 0.1 |
| Scanner | 97.6 ± 0.1 | 97.7 ± 0.2 | 97.2 ± 0.1 | 97.4 ± 0.1 | 99.8 ± 0.0 |
| Pixel spacing | 70.6 ± 0.9 | 69.5 ± 1.0 | 68.8 ± 1.0 | 68.5 ± 1.0 | 94.1 ± 0.2 |
| Hospital ID | 73.5 ± 2.0 | 56.7 ± 1.6 | 54.9 ± 2.4 | 54.5 ± 2.4 | 90.9 ± 1.4 |
| Year of study | 63.9 ± 5.5 | 51.4 ± 4.4 | 51.6 ± 3.4 | 49.4 ± 2.5 | 88.7 ± 1.2 |
| GA | 56.0 ± 1.5 | 50.3 ± 1.8 | 49.4 ± 0.4 | 49.3 ± 0.9 | 82.1 ± 0.6 |
| BMI | 37.2 ± 1.5 | 39.2 ± 1.3 | 37.1 ± 1.4 | 37.5 ± 1.3 | 65.4 ± 0.7 |
| Parity | 58.2 ± 7.5 | 38.0 ± 1.6 | 37.0 ± 1.8 | 35.4 ± 1.4 | 59.2 ± 2.1 |
| Maternal age | 26.9 ± 2.1 | 27.6 ± 1.6 | 27.5 ± 1.5 | 24.9 ± 2.5 | 52.0 ± 1.4 |
| Smoking status | 73.5 ± 8.4 | 51.0 ± 0.5 | 51.2 ± 0.9 | 50.0 ± 1.5 | 52.0 ± 1.0 |
| Ethnicity | 89.3 ± 0.8 | 49.9 ± 1.4 | 50.0 ± 1.6 | 49.8 ± 1.4 | 46.0 ± 3.9 |
| **Cervix dataset** | | | | | |
| Term Birth | 55.7 ± 1.8 | 57.5 ± 2.1 | 57.1 ± 1.6 | 55.4 ± 2.0 | 60.7 ± 2.3 |
| Scanner | 97.4 ± 0.8 | 93.1 ± 3.7 | 85.6 ± 2.5 | 88.6 ± 2.6 | 98.9 ± 0.2 |
| Pixel spacing | 72.7 ± 15.2 | 70.4 ± 16.1 | 70.0 ± 16.4 | 70.1 ± 16.3 | 93.7 ± 4.8 |
| Cervical length | 62.5 ± 7.8 | 65.2 ± 7.0 | 61.8 ± 9.0 | 62.8 ± 8.7 | 85.0 ± 4.9 |
| Hospital ID | 46.3 ± 14.3 | 39.9 ± 14.9 | 38.9 ± 15.3 | 37.9 ± 15.4 | 81.2 ± 9.2 |
| GA | 39.3 ± 4.5 | 32.6 ± 3.8 | 30.6 ± 3.3 | 30.7 ± 3.3 | 66.9 ± 3.2 |
| Year of study | 26.5 ± 3.7 | 25.6 ± 3.0 | 24.2 ± 3.2 | 24.2 ± 3.2 | 62.9 ± 3.7 |
| Birth weight | 21.5 ± 2.2 | 19.7 ± 1.6 | 18.1 ± 0.9 | 16.9 ± 1.2 | 54.2 ± 0.5 |
| BMI | 45.6 ± 4.8 | 24.3 ± 1.2 | 25.0 ± 0.8 | 23.8 ± 0.5 | 51.9 ± 2.0 |
| Maternal Age | 30.7 ± 3.8 | 24.8 ± 5.1 | 26.7 ± 0.9 | 25.1 ± 3.4 | 51.9 ± 0.8 |
| Placenta weight | 35.5 ± 3.2 | 25.2 ± 2.0 | 25.4 ± 2.2 | 24.8 ± 2.1 | 50.5 ± 5.4 |
| Smoking status | 85.4 ± 1.8 | 50.4 ± 1.6 | 50.6 ± 1.7 | 50.4 ± 1.6 | 50.3 ± 2.5 |
| Ethnicity | 89.8 ± 1.5 | 48.9 ± 1.6 | 49.0 ± 1.6 | 48.9 ± 1.6 | 50.3 ± 2.4 |
| Parity group | 61.5 ± 1.1 | 34.1 ± 3.4 | 32.9 ± 1.1 | 31.7 ± 1.7 | 46.4 ± 3.9 |

Table 3: Test performance of ResNet50 classifier model trained to predict various target values in the fetal plane and cervix datasets. Continuous values are discretized and grouped into bins.

| Target | Accuracy | Precision | Recall | F1 | AUROC |
|---|---|---|---|---|---|
| Term Birth | $67.5 \pm 2.5$ | $67.7 \pm 2.4$ | $67.5 \pm 2.5$ | $67.4 \pm 2.6$ | $73.9 \pm 3.0$ |
| Px Spacing | $70.9 \pm 3.8$ | $69.4 \pm 3.1$ | $70.3 \pm 3.3$ | $69.4 \pm 3.1$ | $95.3 \pm 0.9$ |
| Py Spacing | $70.5 \pm 3.0$ | $66.6 \pm 2.9$ | $68.6 \pm 3.1$ | $66.6 \pm 3.1$ | $95.2 \pm 1.0$ |
| Cervical Length | $52.6 \pm 2.0$ | $52.8 \pm 2.3$ | $55.6 \pm 1.9$ | $53.3 \pm 2.0$ | $80.0 \pm 1.6$ |
| Scanner | $56.3 \pm 6.4$ | $32.0 \pm 3.1$ | $55.7 \pm 9.9$ | $30.4 \pm 5.5$ | $77.2 \pm 5.7$ |
| Birth Weight | $21.4 \pm 1.5$ | $22.0 \pm 1.8$ | $23.7 \pm 2.5$ | $21.0 \pm 1.9$ | $61.2 \pm 1.9$ |
| Placenta Weight | $28.8 \pm 1.5$ | $27.5 \pm 1.2$ | $30.0 \pm 3.7$ | $25.0 \pm 1.6$ | $55.4 \pm 2.9$ |
| BMI | $27.4 \pm 2.5$ | $26.9 \pm 2.2$ | $28.4 \pm 3.6$ | $23.1 \pm 2.4$ | $53.3 \pm 3.0$ |
| GA | $34.6 \pm 3.0$ | $27.7 \pm 3.3$ | $27.7 \pm 3.5$ | $27.5 \pm 3.4$ | $48.4 \pm 2.0$ |

Table 4: Test performance of SA-SonoNet model trained for pre-term birth prediction and subsequently fine-tuned to predict the other targets.

**Appendix B. Full covariance matrix plot**

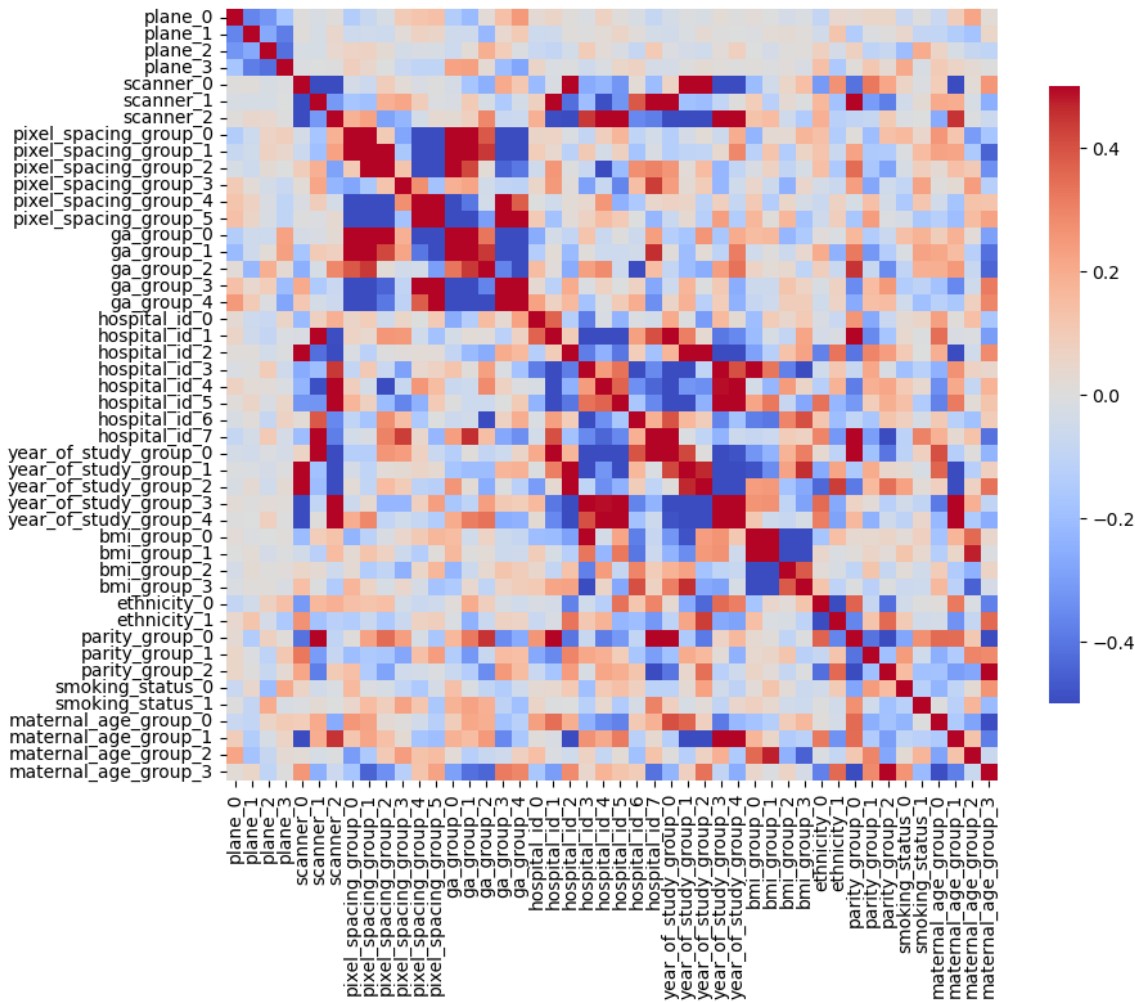

Figure 6: Full correlation matrix between weight vectors from classification head of each targets, extracted from fine-tuned classifier model trained using the entire fetal dataset.

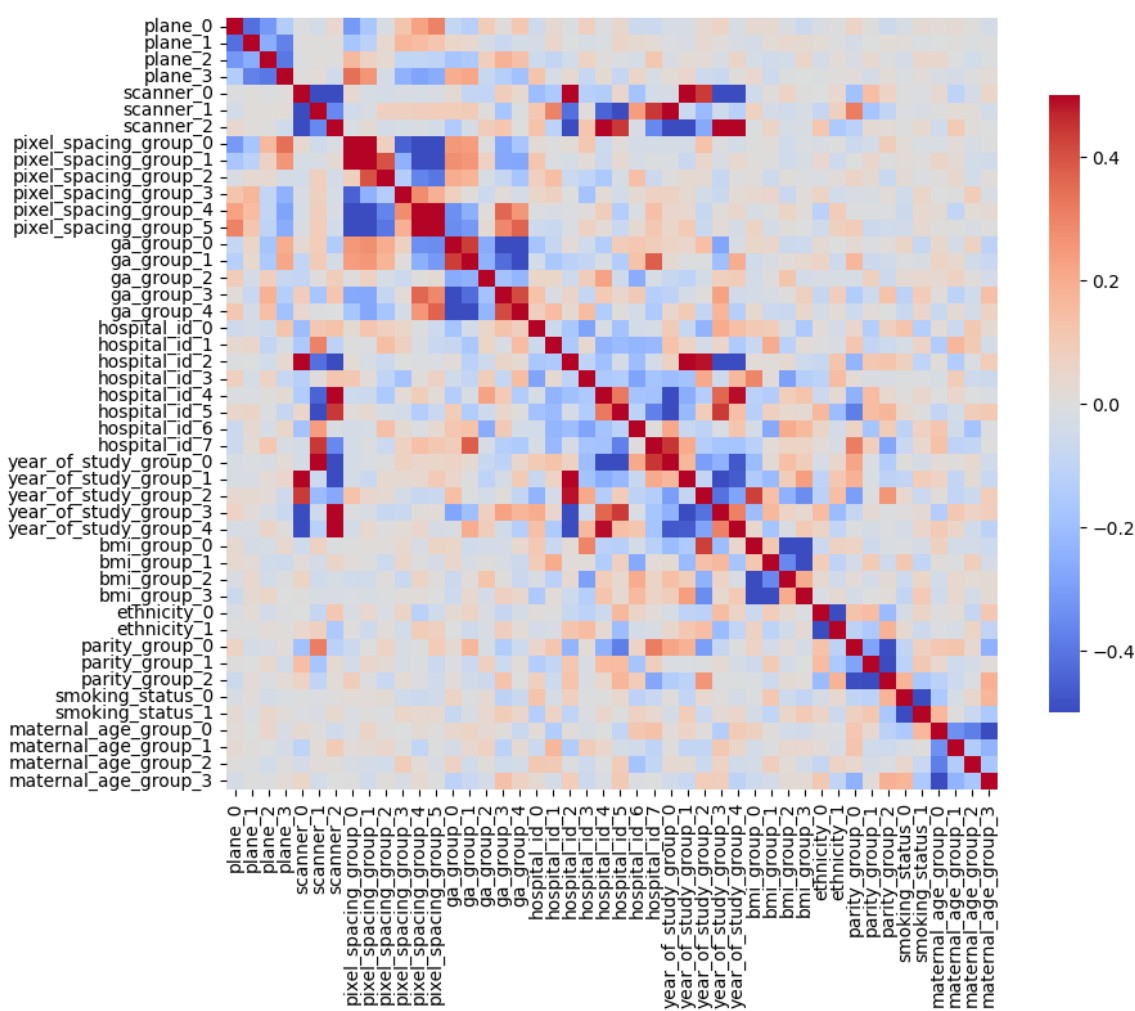

Figure 7: Full correlation matrix between weight vectors from classification head of each targets, extracted from multitask classifier model trained using the entire fetal dataset.

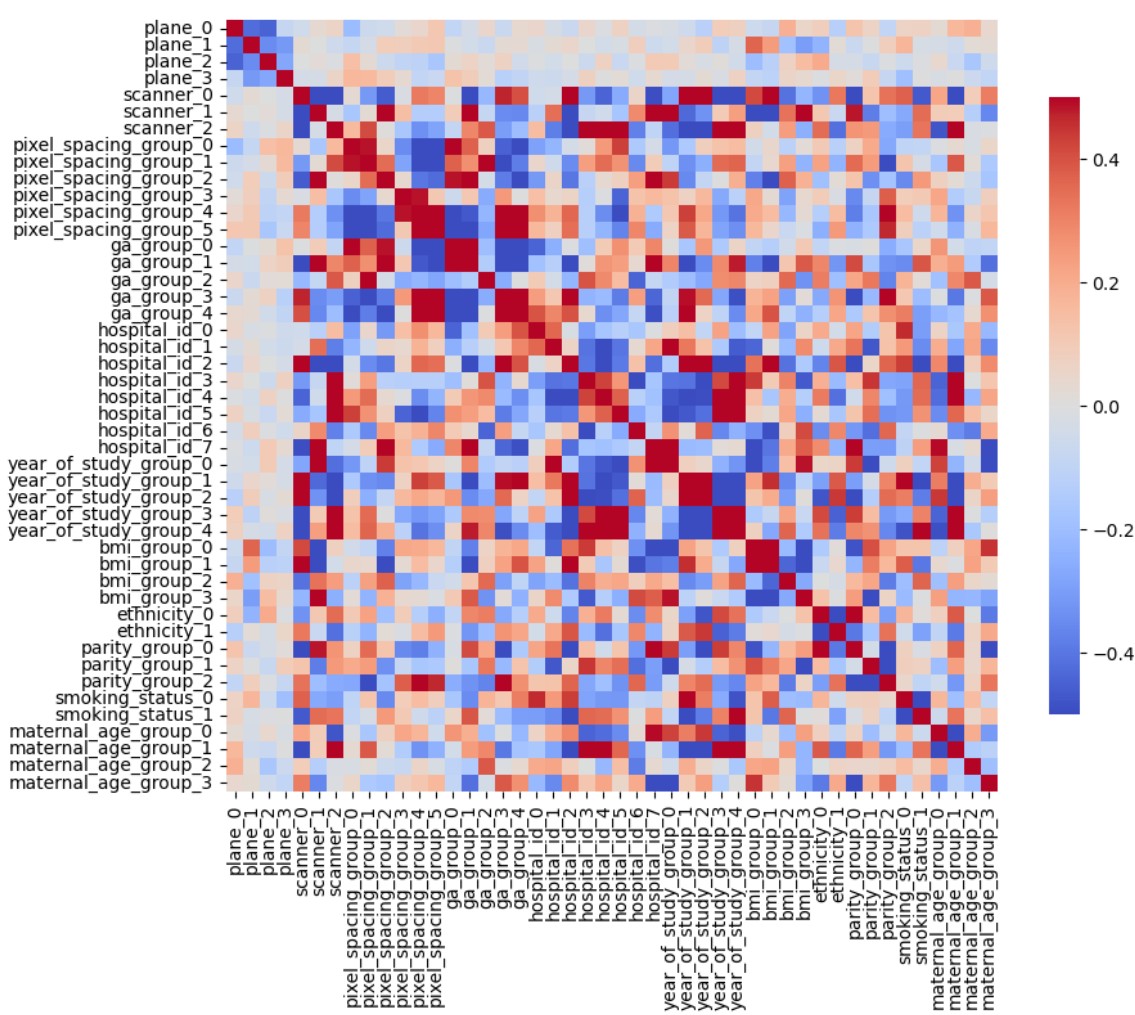

Figure 8: Full correlation matrix between weight vectors from classification head of each targets, extracted from fine-tuned classifier model trained using the balanced fetal dataset.

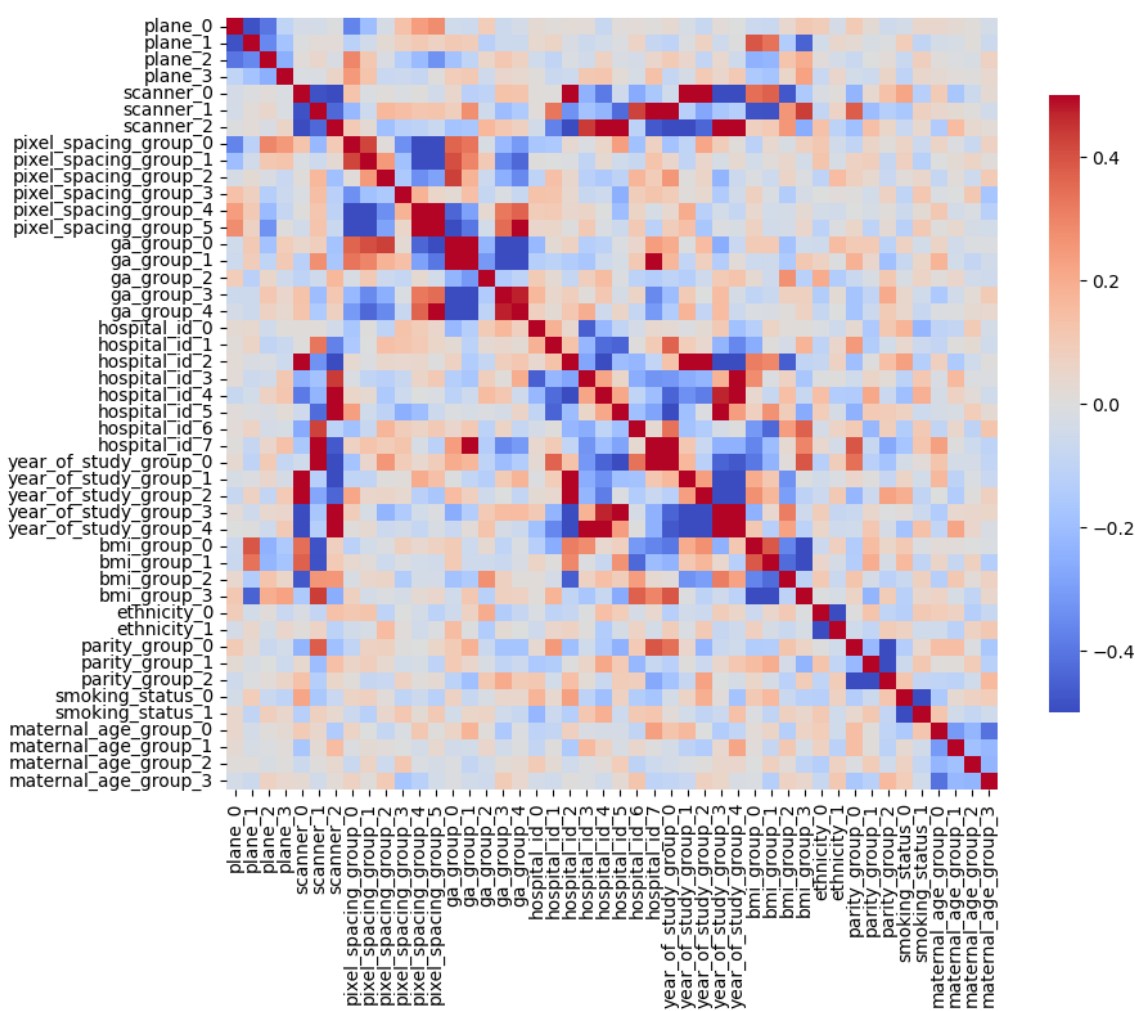

Figure 9: Full correlation matrix between weight vectors from classification head of each targets, extracted from multitask classifier models trained using the balanced fetal dataset.

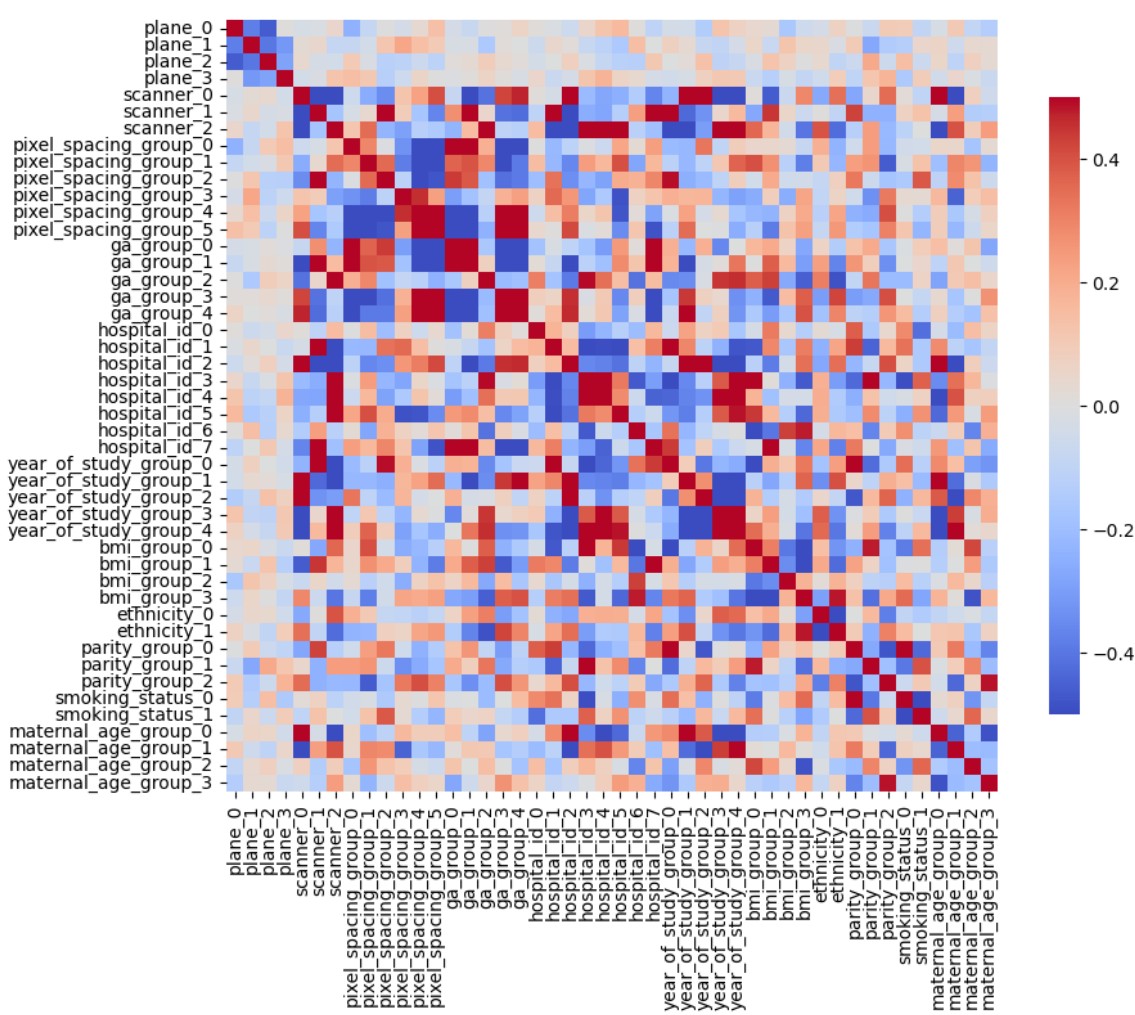

Figure 10: Full correlation matrix between weight vectors from classification head of each targets, extracted from fine-tuned classifier model trained using the fetal dataset with induced bias.

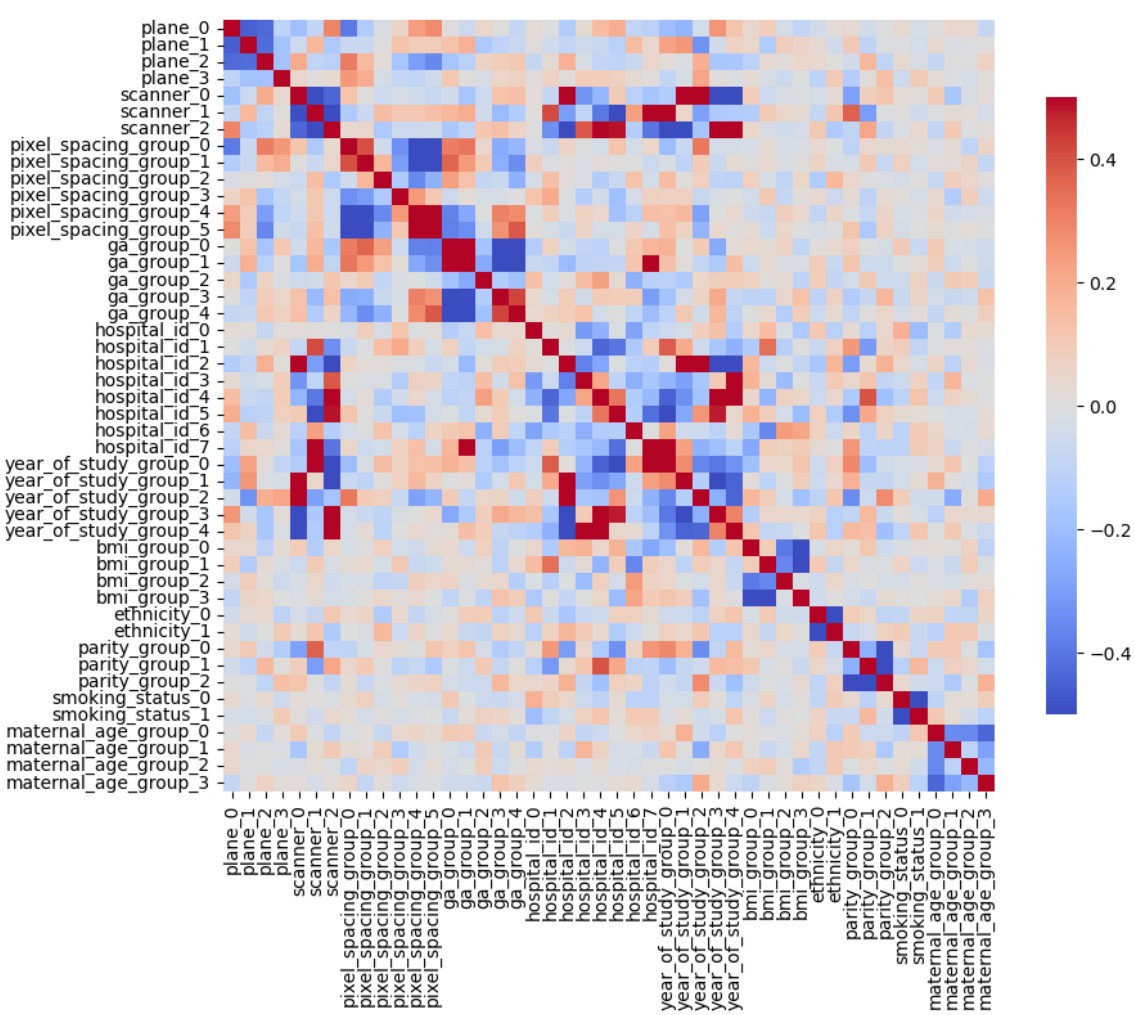

Figure 11: Full correlation matrix between weight vectors from classification head of each targets, extracted from multitask classifier models trained using the fetal dataset with induced bias.

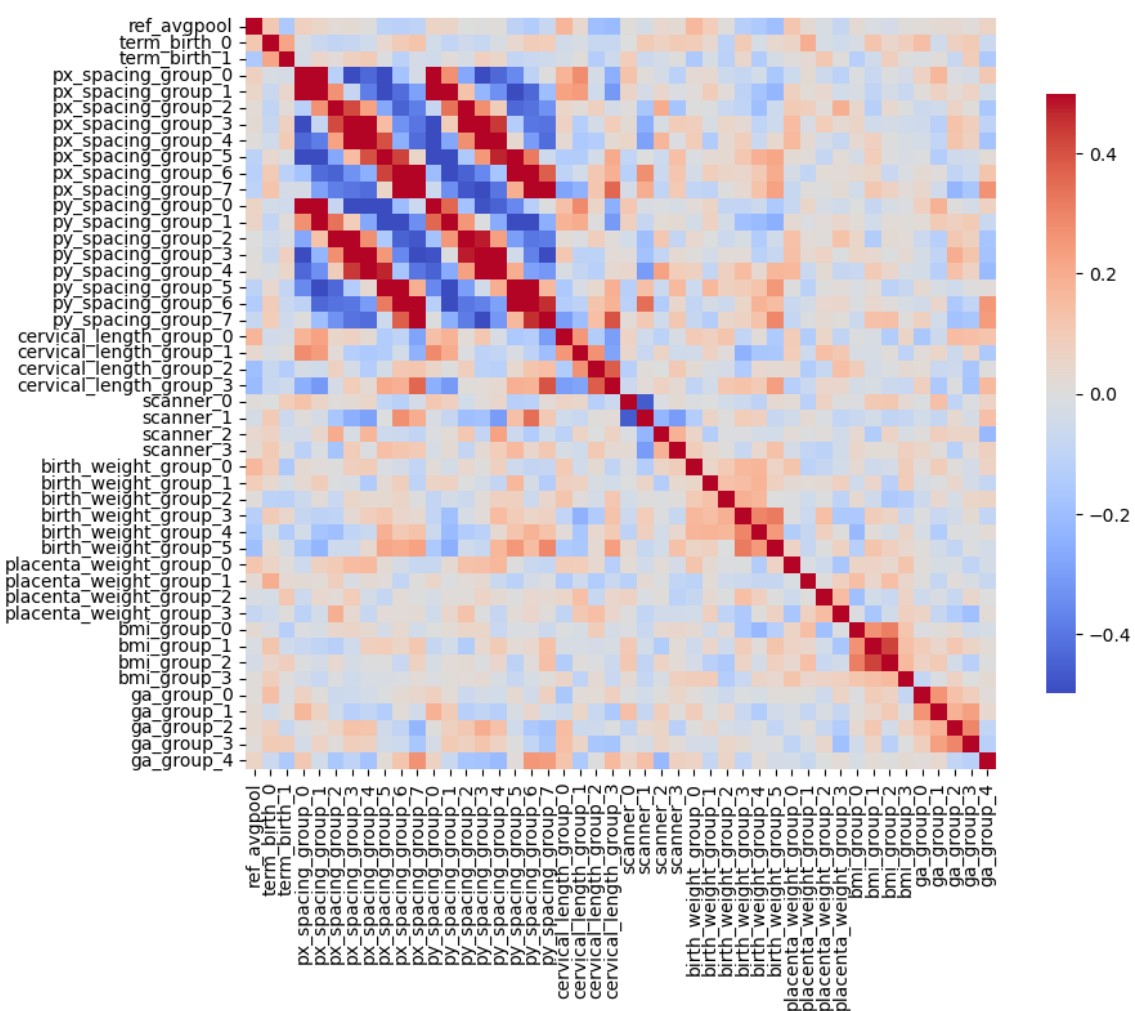

Figure 12: Full correlation between weight vectors from classification head of the targets of the fine-tuned SA-SonoNet model. A reference line is added to represent the average pool layer in the original model.

# Appendix C. Full Results: Validation of Shortcut Learning Detection via Induced Bias

| Target | Fine-tuned model | | | Multitask model | | |
|---|---|---|---|---|---|---|
| | Accuracy | F1 | AUROC | Accuracy | F1 | AUROC |
| Plane | $99.0 \pm 0.6$ | $99.0 \pm 0.6$ | $75.0 \pm 0.0$ | $97.2 \pm 1.1$ | $97.2 \pm 1.1$ | $74.9 \pm 0.0$ |
| Scanner | $77.0 \pm 2.9$ | $76.7 \pm 2.8$ | $91.7 \pm 0.7$ | $95.4 \pm 0.7$ | $95.4 \pm 0.7$ | $99.3 \pm 0.3$ |
| Pixel Spacing | $56.1 \pm 3.5$ | $47.0 \pm 4.2$ | $86.6 \pm 1.0$ | $61.1 \pm 2.4$ | $53.5 \pm 3.5$ | $91.9 \pm 0.8$ |
| Year Of Study | $45.9 \pm 4.5$ | $39.6 \pm 3.4$ | $76.6 \pm 3.1$ | $61.0 \pm 2.2$ | $52.0 \pm 1.6$ | $90.5 \pm 0.6$ |
| GA | $47.2 \pm 3.4$ | $37.0 \pm 2.2$ | $69.8 \pm 1.8$ | $58.0 \pm 1.8$ | $46.6 \pm 1.1$ | $82.8 \pm 0.7$ |
| BMI | $34.9 \pm 3.9$ | $22.8 \pm 3.1$ | $67.3 \pm 1.2$ | $50.0 \pm 2.2$ | $37.9 \pm 5.5$ | $75.9 \pm 2.4$ |
| Hospital ID | $45.3 \pm 0.7$ | $33.2 \pm 2.3$ | $64.6 \pm 1.2$ | $66.8 \pm 1.2$ | $48.3 \pm 1.7$ | $78.0 \pm 0.2$ |
| Parity | $59.0 \pm 10.5$ | $37.2 \pm 3.1$ | $59.8 \pm 2.4$ | $59.0 \pm 7.4$ | $36.3 \pm 2.2$ | $68.2 \pm 2.1$ |
| Smoking Status | $81.6 \pm 6.0$ | $48.2 \pm 1.9$ | $59.8 \pm 2.0$ | $77.7 \pm 6.9$ | $48.5 \pm 3.0$ | $54.3 \pm 4.8$ |
| Maternal Age | $24.0 \pm 2.2$ | $21.7 \pm 2.7$ | $53.2 \pm 1.9$ | $29.5 \pm 5.5$ | $28.7 \pm 4.8$ | $53.9 \pm 3.1$ |
| Ethnicity | $82.8 \pm 10.4$ | $47.0 \pm 1.9$ | $50.1 \pm 8.7$ | $70.9 \pm 12.8$ | $43.9 \pm 3.1$ | $44.6 \pm 6.1$ |
| Term Birth | $67.1 \pm 7.4$ | $42.5 \pm 2.4$ | $35.7 \pm 4.5$ | $67.0 \pm 9.1$ | $47.1 \pm 3.6$ | $46.3 \pm 4.6$ |

Table 5: Test performance of ResNet50 classifier model when fine-tuned to predict the other targets, or trained to predict various targets in a multitask setting, using the balanced fetal dataset.

| Target | Fine-tuned model | | | Multitask model | | |
|---|---|---|---|---|---|---|
| | Accuracy | F1 | AUROC | Accuracy | F1 | AUROC |
| Plane | $98.3 \pm 0.9$ | $98.3 \pm 0.9$ | $75.0 \pm 0.0$ | $95.9 \pm 0.9$ | $95.9 \pm 0.9$ | $74.8 \pm 0.1$ |
| Scanner | $80.0 \pm 2.3$ | $80.0 \pm 2.3$ | $94.2 \pm 1.0$ | $95.6 \pm 0.5$ | $95.6 \pm 0.4$ | $99.2 \pm 0.1$ |
| Pixel Spacing | $51.8 \pm 2.1$ | $45.0 \pm 1.2$ | $85.0 \pm 1.3$ | $64.3 \pm 2.3$ | $61.7 \pm 2.6$ | $91.8 \pm 0.5$ |
| Year Of Study | $51.9 \pm 4.3$ | $40.5 \pm 2.6$ | $82.3 \pm 1.6$ | $67.3 \pm 4.3$ | $50.2 \pm 2.8$ | $91.4 \pm 0.8$ |
| GA | $46.8 \pm 1.6$ | $40.0 \pm 1.9$ | $75.3 \pm 1.1$ | $55.7 \pm 2.5$ | $46.1 \pm 2.4$ | $83.6 \pm 0.5$ |
| Hospital ID | $45.1 \pm 0.7$ | $31.1 \pm 0.6$ | $75.2 \pm 1.3$ | $66.2 \pm 2.6$ | $45.4 \pm 2.9$ | $85.8 \pm 1.3$ |
| Parity | $58.8 \pm 4.0$ | $40.9 \pm 4.4$ | $71.0 \pm 1.3$ | $71.2 \pm 3.5$ | $33.6 \pm 1.5$ | $68.9 \pm 2.7$ |
| Smoking Status | $83.5 \pm 3.5$ | $49.4 \pm 3.0$ | $55.9 \pm 7.9$ | $84.7 \pm 4.2$ | $51.2 \pm 2.8$ | $47.8 \pm 3.5$ |
| Maternal Age | $26.9 \pm 2.3$ | $24.9 \pm 2.7$ | $54.7 \pm 2.8$ | $28.8 \pm 4.9$ | $28.1 \pm 4.0$ | $54.1 \pm 3.4$ |
| Ethnicity | $90.0 \pm 3.1$ | $49.8 \pm 1.8$ | $54.5 \pm 9.8$ | $93.1 \pm 1.4$ | $51.8 \pm 3.2$ | $46.6 \pm 2.6$ |
| Term Birth | $65.5 \pm 5.0$ | $43.6 \pm 1.1$ | $40.0 \pm 1.7$ | $84.4 \pm 0.6$ | $48.2 \pm 0.7$ | $48.2 \pm 3.2$ |
| BMI | $34.0 \pm 2.5$ | $23.1 \pm 1.7$ | $39.1 \pm 1.5$ | $45.6 \pm 6.9$ | $29.9 \pm 3.2$ | $48.9 \pm 1.5$ |

Table 6: Test performance of ResNet50 classifier model when fine-tuned to predict the other targets, or trained to predict various targets in a multitask setting, using the biased fetal dataset.

# Appendix D. Full Results: Empirical Determination of the PCA Projection Threshold

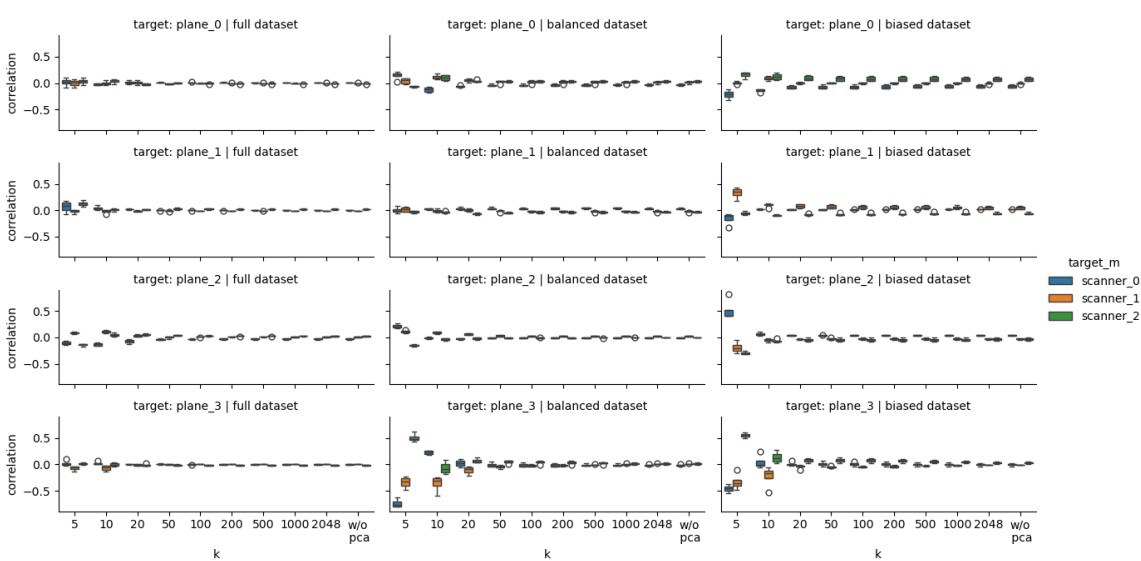

Figure 13: Sensitivity of correlation values to the number of principal components $k$ in fine-tuned models.

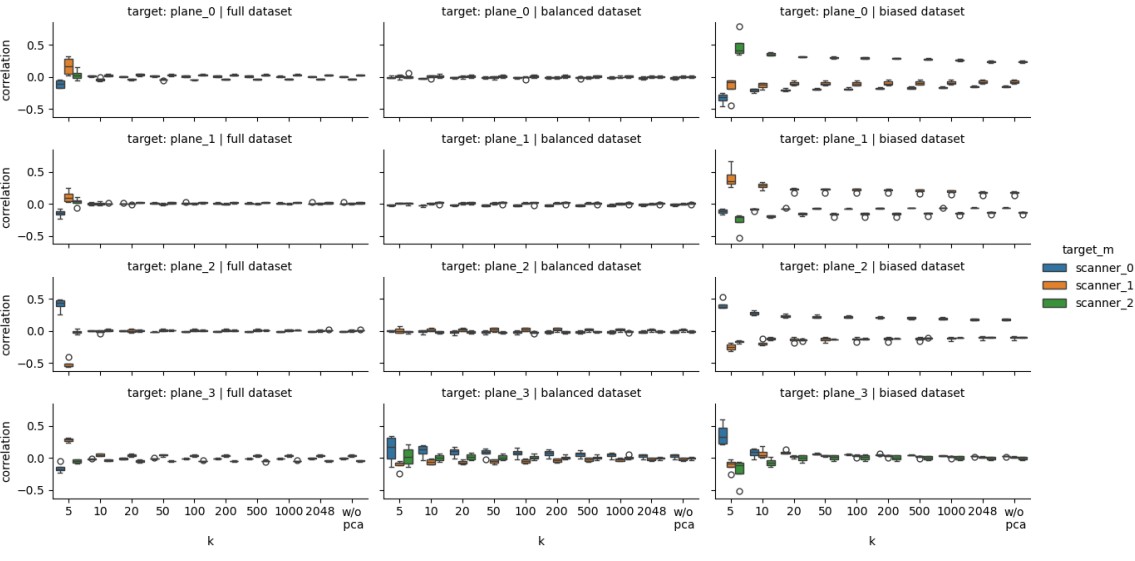

Figure 14: Sensitivity of correlation values to the number of principal components $k$ in multi-task models.

