# OpenReview forum: "Weight Space Correlation Analysis: Quantifying Feature Utilization in Deep Learning Models"
_MIDL.io/2026/Conference — MIDL 2026 Poster_

### Official Review · Reviewer_RHE6 · 2026-01-05

**Confidence:** 5
**Preliminary Rating:** 2
**Final Rating:** 4

**Summary:**

This paper explores a critical challenge in deep learning for medical imaging: the use of 'shortcuts' in model predictions. By employing weight space correlation analysis, the authors investigate how models utilize non-target features. They conducted experiments on two specific datasets, the Fetal and Cervix,and demonstrate that this method can effectively identify whether metadata is being leveraged as a shortcut for predictions.

**Strengths:**

* The manuscript provides a clear motivation and a well-structured narrative.
* Furthermore, the experimental results are supported by evaluations performed on two clinically relevant datasets.
* The experimental framework includes baseline and multi-task configurations, utilizing induced bias to analyze model behavior and shortcut reliance.

**Weaknesses:**

* **Literature Review :** The introduction would benefit significantly from a more comprehensive discussion of existing literature regarding shortcut identification. Specifically, incorporating a discussion of PCA-based quantitative analysis and counterfactual studies would help the reader better position this study within the current state of the art.

* **Methodological Rigor:** The methodological explanations require further detail. In particular, the rationale for the PCA projection of weights needs clarification. A more robust mathematical explanation is necessary to justify why projecting a weight vector onto the feature space of the backbone or data is an appropriate analytical approach.

* **Quantitative Results:** The findings regarding shortcut utilization currently rely on qualitative assessments of color scales. Providing quantitative metrics would strengthen the results and allow for a more objective evaluation of the model's behavior.

* **Support for Claims:** While the authors claim that the induced bias dataset demonstrates that metadata are utilized as shortcuts, this conclusion is not immediately evident from the provided figures. Additional evidence or clearer visualization is needed to support this assertion.

* **Novelty and Comparative Analysis:** Given that this work builds upon the framework established by Glocker et al., the authors should explicitly clarify the advantages of their weight-based correlation analysis compared to the Kolmogorov-Smirnov (K-S) tests in the PCA space used in the original study.

**Detailed Comments:**

**Major Revisions**

Please refer to the "Weaknesses" section above. The primary concerns involve the need for a more rigorous mathematical justification of the weight projection method, a more comprehensive literature review to position the work, and the transition from qualitative visual assessments to quantitative metrics.

**Minor Revisions**

Typography: The quotation marks are inverted (flipped) in several instances throughout the manuscript. Please ensure that opening and closing marks follow standard LaTeX/typographic conventions (e.g., using `` and '').

Visualization of Correlation Matrices: In Figure 1 and Figure 2, the visualization of the results could be significantly improved by removing the self-correlation diagonal (where $r = 1$).

**Justification Of Final Rating:**

I am pleased with the planned edits to the PDF. This research topic is vital to the field of medical imaging, and further discussion is essential to help translate these findings into clinical practice. All the best!

**Justification Of The Preliminary Rating:**

The paper addresses a highly relevant and timely problem in medical imaging: the lack of transparency regarding how deep learning models utilize metadata and shortcuts. The motivation is sound, the manuscript is well-written, and the use of clinically significant datasets (Fetal and Cervix) is a clear strength. However, the current version of the manuscript lacks the methodological depth and quantitative evidence necessary to fully support its conclusions. My preliminary rating is primarily driven by the weaknesses mentioned above.

**Questions To Address In The Rebuttal:**

1. Comparative Advantage: How does the proposed weight-based correlation analysis offer superior diagnostic power compared to the $K$-$S$ tests on the $PCA$ space utilized in the Glocker et al. framework?

2. Could the authors provide quantitative metrics (such as correlation coefficients or $p$-values) to support the claims made regarding the figures? Relying on qualitative color-scale interpretations makes it difficult to assess the effect size of the shortcut utilization.

3. Evidence of Induced Bias: could the authors provide a more granular breakdown or an alternative visualization that explicitly demonstrates the model's reliance on metadata? Currently, it is difficult to definitively conclude from the figures that metadata is the primary driver of the observed shortcuts.

---

> ### Author Response · Authors · 2026-01-24
>
> We thank the reviewer for their rigorous assessment. We appreciate the reviewer's high confidence and the expertise reflected in their critique. Below are our responses to their specific points:
> - Methodological Rigor and Mathematical Justification: We have completely restructured the Materials and Methods section (Section 2). We now provide a formal mathematical derivation for the weight projection, defining the linear classification head as an "attention vector" $w_i \in \mathbb{R}^d$ that operates on the latent embedding $z$. We justify the PCA projection $P \in \mathbb{R}^{k \times d}$ as a mapping onto the empirical data manifold, ensuring that the resulting cosine similarities $\cos_{ij}$ are computed only along directions of active variance where the model actually encounters data, rather than in the sparse, orthogonal regions of the high-dimensional embedding space.
>
> - Comparative Advantage vs. Glocker et al.: We have expanded the Introduction to clarify the distinction between our work and the foundational work of Glocker et al. (2023). While Glocker et al. use Kolmogorov-Smirnov (K-S) tests to quantifying the separation of embeddings relative to ground-truth metadata labels, their approach is an unsupervised exploration of the embeddings. Our method focuses on the classification head. The critical distinction lies in the transition from representation to decision: while Glocker et al. demonstrate that inputs with differing ground truth metadata are distinctly clustered or correlated within the embedding space, WSC analysis determines whether the model actually leverages those latent structures. In other words, while they confirm the presence of confounding information in the embeddings, our method confirms its active utilization by the classification head.
>
> - Quantitative Evidence and Granular Visualization: We agree that the previous heatmaps relied too heavily on qualitative color-scale interpretation. We have addressed this in three ways:
>   - Improved Visualization: We have updated the color scales in the heatmaps to cap at lower correlation values.
>   - Tabulated Significance: While a $50 \times 50$ table is unreadable, we have added a boxplot visualization in the new Section 3.5 (Sensitivity Analysis). This plot shows the distribution of correlation values across multiple seeds, clearly demonstrating the statistical shift in correlation when moving from a balanced dataset to a biased one.
>   - Null Distribution: We have added a section establishing the null distribution of correlation values (Section 3.2.1), providing a baseline that allows the reader to judge the effect size of observed shortcuts against random chance.
>
> - Evidence of Induced Bias: To provide more definitive proof of the model's reliance on metadata, we added a control group: the Induced Balanced Dataset. By showing that WSC values remains at the null baseline when plane and scanner are uncorrelated—even though the scanner information is still "encodable"—we provide the necessary "negative control" to support our claims about the Biased dataset.
>
> - Literature Review and Typographic Standards: We have conducted a more comprehensive literature review, incorporating recent research to better position WSC analysis. Additionally, we have also corrected the inverted quotation marks and other LaTeX typographic errors throughout the manuscript.

---

### Official Review · Reviewer_PjDp · 2026-01-07

**Confidence:** 3
**Preliminary Rating:** 3
**Final Rating:** 5

**Summary:**

The paper introduces a novel method, Weight Space Correlation Analysis (WSCA), for detecting and measuring shortcut learning. While most studies focus on detecting metadata signals in embeddings, the authors address whether a model actually uses these metadata signals (e.g., scanner/site) for clinical prediction, rather than merely encoding them in its representations. The method relies on metadata heads, training linear probes on frozen embeddings to predict metadata, projecting layer weight vectors into a PCA subspace of the embedding manifold, and computing cosine similarities between primary-task head weights and metadata-task head weights as a proxy for shared feature utilization. The authors evaluate the framework on two tasks: fetal ultrasound plane classification and preterm birth prediction on the SA-SonoNet dataset to assess relationships between prediction and various clinical/technical factors. The first experiment serves as a proof-of-concept with two controlled scenarios: i) scanner is encoded but not used, and ii) an induced-bias variant showing higher WSCA when plane–scanner correlation is introduced.

**Strengths:**

* The paper addresses an important and practical research gap in shortcut-learning analysis: distinguishing situations in which a model can predict metadata from embeddings from situations in which the model relies on metadata for the main task, which is highly relevant in medical imaging.
* The proposed method is operationally simple and yields intuitive visualizations.
* Experiments include a reasonably controlled validation where induced bias via data curation increases the measured correlations, emphasizing the potential of the proposed method for detecting and measuring shortcut learning.

**Weaknesses:**

* The methodological novelty is moderate. WSCA largely reuses familiar building blocks (linear probing, multi-task heads, and weight/representation geometry) into a diagnostic measure. While the framing of “utilization vs. encodability” is clear, it reads more like an incremental analysis lens than a fundamentally new learning framework or substantially new interpretability method.
* The main point of the method is to treat weight-vector alignment between task- and metadata linear heads as evidence of confounder utilization. Such a hypothesis is plausible but not guaranteed: two heads can be related without weight alignment (representation can rotate/redistribute features), alignment can happen because both tasks correlate with the same true factor (or because the tasks predict each other under bias).
* For softmax classifiers, pairwise decision boundaries depend on (wᵢ − wⱼ), not raw wᵢ alone. Correlating raw class weight vectors can be misleading due to invariances.
* For the PCA, retaining 99% variance (with a floor of 50 PCs) may discard low-variance directions that still drive classification (common in shortcut settings). How do the results vary with changes in the number of PCs or the percentage of variance retained?
* The paper would benefit from a clearer discussion of limitations/failure modes. For instance, would it be possible to observe cases where metadata is encoded, WSCA is low, yet we still observe performance collapse during scanner shifts? What should we conclude?

**Detailed Comments:**

See weaknesses.

**Justification Of Final Rating:**

The authors methodically addressed all my concerns, added the aforementioned experiments, and strengthened the rationale for their method. I therefore recommend acceptance and would like to warmly thank the authors for this contribution.

**Justification Of The Preliminary Rating:**

The paper addresses a highly relevant problem: diagnosing shortcut learning and distinguishing encodability from the utilization of confounders. The authors propose a simple, implementable diagnostic (WSCA) with intuitive visualizations and a reasonable sanity check for induced bias. This makes the work potentially useful for practitioners working with multi-site / multi-scanner medical imaging.

However, the current evidence does not yet justify the central interpretation that WSCA reliably reflects the actual reliance on confounders. The method is only weakly tied to interventional or distribution-shift outcomes (e.g., scanner-held-out generalization or balanced-scanner evaluation), and the heatmaps remain largely qualitative due to missing calibration (null distributions, seed variability, and guidance on what constitutes a meaningful correlation). In addition, the approach may be fragile to implementation choices (PCA, multi-class head parameterization). I therefore rate this paper as borderline, adding one decisive experiment that links WSCA to performance vulnerability, plus basic calibration and a brief PCA sensitivity check, would likely move this to Weak Accept.

**Questions To Address In The Rebuttal:**

1/ Add experiments whose results show that WSCA links to real shortcut reliance. For instance, perform an experiment on a balanced-scanner set (each class has the same number of each scanner) in which we would expect high encoding but low WSCA.

2/ Provide a null WSCA distribution, allowing users to have an idea of when WSCA becomes significant and truly leads to shortcut learning.

3/ Add experiments on robustness to the PCA/implementation choices.

---

> ### Author Response · Authors · 2026-01-24
>
> We thank the reviewer for their constructive critique. We appreciate the recognition of our work as a highly relevant and potentially useful diagnostic for practitioners. We have substantially updated the manuscript to address their concerns regarding the reliability and interpretation of WSC analysis. Below are our responses to the specific points raised:
> - Methodological Novelty and Framing: we agree that WSC analysis utilizes established components. However, we do not claim to introduce a new learning framework, but rather a systematic diagnostic lens to distinguish encoding from utilization. To the best of our knowledge, we are the first to propose and validate a framework that quantitatively compares the "attention" of classification weights to audit model reliance on non-causal factors.
> - Weight Alignment vs. Confounder Utilization: The reviewer had raised two excellent theoretical concerns regarding the link between alignment and utilization:
>   - Feature Redistribution: The reviewer noted that tasks could be related without weight alignment if features rotate. However, for this to occur while maintaining performance, the backbone would need to support separate, non-linear manifolds for each task. If the model relies on the same features at the linear layer, the weights must align. We have clarified in the text that WSC analysis is intended to be used when probing performance is high; if the probe fails, the alignment is indeed meaningless.
>   - True Factor Correlation vs. Shortcut: The reviewer noted that alignment can occur when tasks naturally correlate with a true clinical factor. We argue this is a feature of our method, not a bug. In our analysis of SA-SonoNet, we found moderately high correlation between sPTB and cervical length. This is clinically expected (cervical length is a marker used clinically for assessing risk of sPTB) and does not constitute a "shortcut" in the pejorative sense, but rather a validation that the model leverages physiologically relevant markers rather than acquisition artifacts like scanner model.
> - Pairwise Decision Boundaries ($w_i - w_j$): While the decision boundary is indeed defined by the difference between vectors, our research question is fundamentally about feature selection: "Which regions of the embedding space does a specific class head attend to?" By correlating raw class weights, we isolate the specific directions in the latent space that pull the model toward a specific prediction. This provides a direct measure of whether the "concept" of a specific scanner aligns with the "concept" of a specific clinical anatomy in the model's logic.
>
> - New Experiments and Calibration: We have performed the experiments requested by the reviewer:
>   - Induced Balanced vs. Biased (Calibration): We modified our validation experiment to include a balanced sub-dataset (where each plane has an equal number of scanner samples). As expected, this showed high encoding (scanner could still be predicted) but low WSC values, contrasting with the Induced Bias scenario.
>   - Null Distribution: We have added a section establishing the null distribution of WSC values by computing correlations between unrelated tasks across baseline models. This provides the "calibration" requested to determine when a correlation value becomes statistically significant.
> - PCA Sensitivity Analysis: We added a new section (Section 3.5) studying the robustness of WSC values to the choice of PCA rank $k$. We empirically found that $k \ge 50$ (99% variance) provides a stable plateau, while excessively high $k$ values dilute the signal with orthogonal noise.
> - Limitations and Failure Modes: We have added a Limitations section discussing the "performance collapse" scenario the reviewer has mentioned. We clarify that if a model fails during a scanner shift despite low WSC values, it allows practitioners to rule out shortcut learning and instead investigate other failure modes such as poor image quality or domain shift in the underlying anatomy.

---

### Official Review · Reviewer_1p9r · 2026-01-09

**Confidence:** 3
**Preliminary Rating:** 4

**Summary:**

The authors propose Weight Space Correlation Analysis (WSCA), an interpretable methodology to quantify whether a deep learning model actively utilizes specific features (such as confounding metadata) for its primary classification task, rather than merely encoding them in the latent space. The method involves projecting the weight vectors of the primary task and fine-tuned probes for metadata tasks into a PCA-reduced latent space and computing their cosine similarity to measure alignment. The study validates this approach using two clinical ultrasound datasets (Fetal and Cervix). They demonstrate that while metadata (e.g., scanner ID) is encoded in the embeddings, the model does not necessarily use it for clinical prediction unless a strong bias is artificially induced. Finally, they apply WSCA to the SA-SonoNet model for Spontaneous Preterm Birth (sPTB) prediction, showing it aligns with clinical factors (e.g., birth weight) rather than acquisition artifacts.

**Strengths:**

Relevance: The paper addresses a critical issue in medical image analysis: "shortcut learning" and the "black box" nature of deep learning. Distinguishing between encoding a feature (presence) and utilizing a feature (reliance) is a significant conceptual step beyond standard probing techniques.

Methodological Clarity: The proposed WSCA method is mathematically grounded yet intuitive. Using PCA to define the data manifold ensures that the correlation analysis respects the actual distribution of the data (the directions of variance that actually exist), rather than theoretical high-dimensional orthogonality.

Validation Strategy: The experimental design is sound. It systematically moves from establishing the baseline (proving metadata is encoded via probing) to a "stress test" (multi-task learning) and finally a "validity check" (induced bias). The induced bias experiment (Table 2 and Fig 1c/d) provides strong empirical evidence that the metric behaves as expected when shortcuts are forced.

Real-World Application: Applying the method to a complex, pre-existing model (SA-SonoNet) demonstrates practical utility beyond toy examples. The findings that sPTB prediction correlates with birth weight but not scanner ID adds a layer of trust to the clinical model and illustrates how this tool can be used for model auditing.

**Weaknesses:**

Linearity Assumption: The method relies on analyzing the weights of the final fully connected layer. This assumes that the relationship between the latent features and the class logits is linear. While this is true for the final layer of ResNet-like architectures, it simplifies the complex non-linear feature extraction that happens prior. It detects if the final decision is based on linear combinations of shared features, but might miss deeper, non-linear dependencies if the "shortcut" is entangled earlier in the network or if the classification head is more complex.

Dimensionality Reduction: The reliance on PCA (step 2 in Section 2.4) to define the "data manifold" is crucial. While the authors state they retain 99% variance or a floor of 50 components, the sensitivity of the correlation results to the number of components is not explored. For instance, if the confounder lies in the "tail" of the variance (which might be discarded by PCA in some settings, though likely not here with 99% retention), the method might miss it. A brief sensitivity analysis would strengthen the method's robustness claims. Though perhaps out of scope for a short paper.

Correlation Interpretation: The paper notes that high correlation implies the tasks "look at similar features." However, cosine similarity in weight space aggregates all dimensions. It might be theoretically possible for weights to be orthogonal in critical dimensions but correlated in noise dimensions, or vice versa, though the PCA step mitigates this. More granular visualization or analysis of which specific principal components drive the correlation would be insightful, though again perhaps out of scope for a short paper.

**Detailed Comments:**

Minor Typos:
- Section 1: "One of the primary threat" -> "One of the primary threats".
- Section 3.2.1: "table table 1" -> "Table 1".

**Justification Of The Preliminary Rating:**

The paper presents a novel, methodologically sound, and clinically relevant approach to a significant problem in medical AI: distinguishing feature presence from feature utilization. The distinction is vital for trust. The validation through induced bias is convincing and the application to SA-SonoNet shows real-world promise. The weaknesses regarding linearity assumptions and sensitivity to PCA parameters are notable but do not fundamentally undermine the value of the contribution. It offers a practical tool for auditing model trustworthiness.

**Questions To Address In The Rebuttal:**

Sensitivity Analysis: How sensitive is the Weight Space Correlation (WSC) score to the choice of PCA components? Does the conclusion change if you retain only 90% variance or use a fixed number of components (e.g., 20 vs 100)?

Orthogonality vs. Independence: Can you elaborate on the distinction between weight orthogonality and statistical independence? Is it possible for weights to be orthogonal (low cosine similarity) but for the outputs to still be statistically dependent due to the distribution (covariance) of the embeddings X, considering that PCA projection aligns the basis but does not necessarily whiten the data?

---

> ### Author Response · Authors · 2026-01-24
>
> We thank the reviewer for their thoughtful and constructive feedback. We appreciate the recognition of the conceptual significance of distinguishing feature encoding from feature utilization, as well as the validity of our experimental design. We have carefully addressed the comments in the revised manuscript. Below are our specific responses to the points raised:
> - Sensitivity Analysis and Dimensionality Reduction: We agree that the choice of the PCA rank $k$ is a critical parameter. To address this, we have added a new subsection (Section 3.5) and a corresponding figure (Figure 3) to the manuscript. We conducted a sensitivity analysis by varying $k$ from 10 to 2048 (the full ResNet50 embedding dimension). Our results demonstrate two key phenomena:
>   - Instability at low $k$: For $k < 50$, the mean correlation value fails to reach a stable plateau and exhibits high variance across different random initialization seeds.
>   - Signal dilution at high $k$: As $k$ approaches the full dimensionality, the correlation values decrease significantly.
> This empirically justifies our strategy of maintaining a floor of 50 components while ensuring 99% of the variance is explained. This approach maximizes stability while preventing noise in the "tail" of the variance from diluting the correlation signal.
> - Linearity Assumption: The reviewer correctly identifies that our method focuses on the linear classification head. We have added a Limitations section to the paper to discuss this explicitly. We argue that this assumption is both practical and mathematically motivated:
>   - Ubiquity: The "frozen backbone + linear probe" paradigm is the standard auditing tool in the current literature.
>   - Manifold Linearization: As noted by Bengio et al., a primary goal of deep representation learning is to transform complex, non-linear input manifolds into a space where the data is linearly separable. While a non-linear head might capture deeper dependencies, a linear analysis of the weight space provides a clear, interpretable proxy for the features the model has prioritized during the final decision-making stage.
> - Correlation Interpretation and PCA Rationale: We agree that cosine similarity in raw high-dimensional space can be misleading because it aggregates all dimensions indiscriminately, including those containing only noise. This is precisely why we proposed the PCA projection as a component of WSC analysis. Projecting weights onto the principal components of the data manifold ensures, to a certain extent, that the correlation is computed only along directions where data actually exists. Our sensitivity analysis (Section 3.5) confirms that without this informed projection (i.e., at very high $k$), the correlation can be diluted by noise dimensions, potentially leading to an underestimation of shortcut reliance.
> - Orthogonality vs. Independence: The reviewer raises a profound point regarding whether weight orthogonality (low cosine similarity) necessarily implies statistical independence of the model's outputs. This question touches on the core distinction between our work and the framework established by Glocker et al. (2023). In our framework, weight orthogonality implies that the classification head is not "looking" for the same features that define the metadata task. Statistical independence of the outputs, however, depends on both the weights and the distribution of the embeddings. Even if weight vectors are orthogonal, the outputs could be dependent if the embeddings themselves have high covariance (i.e., the data manifold is biased). We have added a detailed discussion in the Introduction to explicitly distinguish our methodology from that of Glocker et al. While their work is vital for identifying when confounding information is correlated in the latent space, our WSC analysis framework provides the necessary next step: determining whether the classification head's decision boundary actively exploits that information or remains orthogonal to it.
> - Minor Typos: We thank the reviewer for their careful eye. The identified typos in Section 1 and Section 3.2.1 have been corrected.

---

### Author Rebuttal · Authors · 2026-01-24

**Rebuttal:**

We thank all reviewers for their time and their insightful feedback, which has allowed us to significantly strengthen our manuscript. We have addressed all major concerns through extensive revisions and additional experiments:
- Contextual Positioning: Revamped the introduction and literature review to clarify our advantage over other methods, including Glocker et al..
- Theoretical Grounding: Provided a formal mathematical derivation of the weight projection method and a dedicated Limitations section discussing the linear head assumption and failure modes.
- Quantitative Rigor: Added a new subsection (Section 3.2.1) to establish a null distribution of weight correlations to provide a baseline for significance and added boxplots to demonstrate statistical shifts under biased conditions.
- Empirical Validation: Expanded the "Induced Bias" experiment to include an Induced Balanced control group, providing empirical proof that WSC analysis distinguishes between feature encoding and active utilization.
- Sensitivity Analysis: Added a new subsection (Section 3.5) studying the impact of the PCA rank $k$ on correlation values, justifying our strategy ($k \ge 50, 99\%$ variance) to maximize signal-to-noise.
- Typographic/Visual Polish: Corrected multiple typo and improved heatmap color scales for better clarity.

We address each reviewer's comments separately in the allocated sections.

**Supporting Material:**

/attachment/0fc4240d8b1e5fcb0cf70a72aba9eb0911722854.zip

---

### Comment · Area_Chair_RiVr · 2026-02-01
**Updated ratings**

I would like to ask all reviewers to provide their final rating of the paper by factoring in the rebuttal provided by the authors.

Thanks!

---

### Meta-Review · Area_Chair_RiVr · 2026-02-09

**Recommendation:** Accept (Oral)
**Confidence:** 5

**Metareview:**

After the rebuttal phase, all reviewers are in agreement that this contribution is a useful addition to the fairness and shortcut learning literature and that it aims at solving a relevant problem (does shortcut learning actually appear?). I agree with this overall assessment and also agree that the methodological novelty is rather moderate, but that does not undermine its utility for the community.

One additional suggestion that I have: I think it would be worthwhile to also discuss the approach in the light of Stanley et al. (https://doi.org/10.1016/j.ebiom.2024.105501) who also build upon the framework established by Glocker et al., but (also) go beyond that by individually analyzing the PCs in activation space via logistic regression wrt clinical task as well as the auxiliary task. While not as elegantly carried out as in this submission, I think that might be methodologically related.

---

### Decision · Program_Chairs · 2026-02-13

Accept (Poster)